# A control oriented strategy of disruption prediction to avoid the configuration collapse of tokamak reactors

Andrea Murari [1,2,121], Riccardo Rossi [3,121], Teddy Craciunescu [4], Jesús Vega [5], JET Contributors* & Michela Gelfusa [3] ✉

The objective of thermonuclear fusion consists of producing electricity from the coalescence of light nuclei in high temperature plasmas. The most promising route to fusion envisages the confinement of such plasmas with magnetic fields, whose most studied configuration is the tokamak. Disruptions are catastrophic collapses affecting all tokamak devices and one of the main potential showstoppers on the route to a commercial reactor. In this work we report how, deploying innovative analysis methods on thousands of JET experiments covering the isotopic compositions from hydrogen to full tritium and including the major D-T campaign, the nature of the various forms of collapse is investigated in all phases of the discharges. An original approach to proximity detection has been developed, which allows determining both the probability of and the time interval remaining before an incoming disruption, with adaptive, from scratch, real time compatible techniques. The results indicate that physics based prediction and control tools can be developed, to deploy realistic strategies of disruption avoidance and prevention, meeting the requirements of the next generation of devices.

The increasing acceleration of climate change and its consequences has recently emphasised the need for humanity to find clean sources of energy and in particular of electricity[1]. In the medium and long term, thermonuclear fusion could potentially become a very important ingredient in a sustainable energy mix on a planetary scale[2]. This approach to electricity generation is based on coalescing the nuclei of hydrogen isotopes; the resulting defect of mass translates into very efficient energy production[3]. Indeed, nuclear fusion is the most exoenergetic reaction in the known universe and it is the power sources of stars such as our sun.

On earth, the most practical reaction is the one between two hydrogen isotopes: deuterium (D) and tritium (T). However, their nuclei, in order to fuse, need to overcome the repulsive Coulomb barrier and come very close, to distances of the same order as the

nuclear radius. Unfortunately this is not easy to achieve in the laboratory. The most advanced alternative consists of heating the fuel, to produce the fourth state of matter in the form of very high temperature plasmas. These plasmas, routinely reaching temperatures higher than the core of the sun, cannot come in contact with material surfaces and must be confined by other means. One potential solution is based on containing them with very strong magnetic fields. This is the approach called Magnetic Confinement Nuclear Fusion (MCNF)[4].

The tokamak remains the most realistic magnetic configuration in the perspective of developing a commercially viable thermonuclear fusion reactor[5]. The topology of the magnetic fields in a tokamak is shown graphically in Fig. 1a. Unfortunately, the configuration is affected by macroscopic instabilities, which are called disruptions[6,7] and can cause the abrupt extinction of the plasma

[1]Consorzio RFX (CNR, ENEA, INFN, Università di Padova, Acciaierie Venete SpA), Corso Stati Uniti 4, Padova, Italy. [2]Istituto per la Scienza e la Tecnologia dei Plasmi, CNR, Padova, Italy. [3]University of Rome "Tor Vergata", via del Politecnico 1, Roma, Italy. [4]National Institute for Laser, Plasma and Radiation Physics, Magurele-Bucharest, Romania. [5]Laboratorio Nacional de Fusión, CIEMAT. Av. Complutense 40, Madrid, Spain. [121]These authors contributed equally: Andrea Murari, Riccardo Rossi. *A list of authors and their affiliations appears at the end of the paper. ✉e-mail: gelfusa@ing.uniroma2.it

(a)

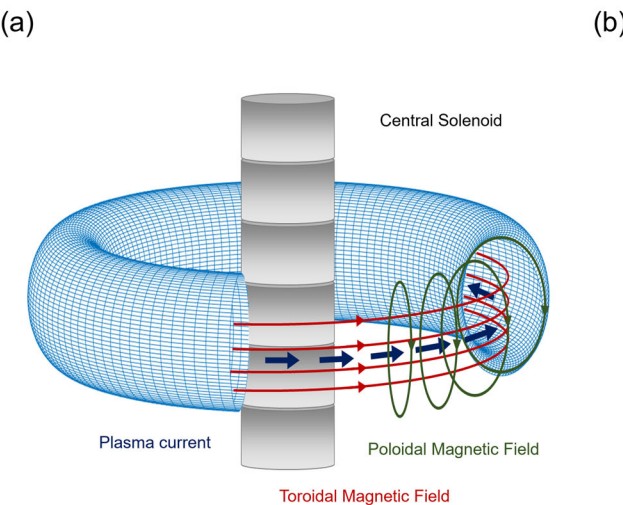

(b)

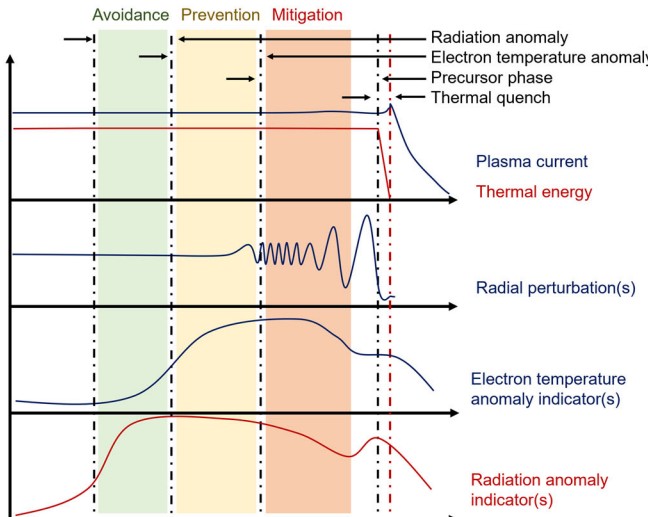

**Fig. 1 | Magnetic topology and disruption precursors. a** Topology of the magnetic fields in a tokamak. **b** Time evolution of typical disruption precursors during normal operation and in the phase leading to a disruption. In agreement with the literature, the beginning of the current quench is considered the disruption time. Avoidance consists of the remedial actions, allowing to recover a healthy plasma state to continue the experiments. Prevention measures are meant to terminate the discharge quickly, before the actual occurrence of the disruption. When a disruption is unavoidable, because there is no time for avoidance or prevention, mitigating its consequences with appropriate tools, such as shattered pellets or massive gas injection, is the only remaining option[7].

current, with potential disastrous consequences for future large machines, such as ITER (the international tokamak under construction in the south of France) and DEMO (the demonstrative reactor)[8,9]. The qualitative evolution of the main plasma quantities, in the phase approaching and during a disruption, are reported in Fig. 1b. The disruption proper start with a thermal quench, in which almost all the thermal energy of the plasma is lost to the wall on time scales of milliseconds. In future devices, such as DEMO, in this phase the energy loads on the plasma facing components could cause the melting of even the most resistant present-day materials. The plasma current decays on much longer time scales, during the so called current quench. Also this phase can be very dangerous though, due to the ensuing induced electromagnetic loads. In ITER, the forces on the metallic structures surrounding the plasma could be higher than the weight of an Airbus A380 plane. Consequently, in reactor scale devices it will be imperative to limit the number of disruptions, not only to guarantee continuity of operation but also to preserve the structural integrity of the machines. Fortunately, several signals show an anomalous behaviour in the phases leading to the beginning of the current quench. Detecting these precursors and interpreting them, in order to provide the control system with clear indications about the criticality of the plasma state, is the objective of disruption predictors (see section 'Methods').

To reduce the tritium inventory in their vacuum vessels, the next generation of devices will be equipped with metallic plasma facing components. The Joint European Tokamak (JET) is one of the largest tokamaks ever operated in the world and it is also fitted with a metallic wall: Be in the main chamber and W in the divertor[10]. JET is also the only present device with the capability of using the fuel of the reactor. In the second half of 2021, the most ambitious series of experiments in the world, scanning systematically the isotopic composition from pure hydrogen to full tritium and 50-50 DT, was carried out on JET[11].

Unfortunately, operation with metallic walls is proving particularly vulnerable to disruptions also in present-day devices. On JET at high currents and powers, the disruptivity of both the baseline and the hybrid scenarios exceeds by about an order of magnitude the maximum levels admitted in ITER at target values, let alone DEMO, as shown in Table 1[12]. This a consequence of the fact that disruptions are a very complex and nonlinear phenomenon, involving several entangled effects: the peaking of the density profile, the hollowing of the temperature profile, the transport of impurities, the pedestal properties, centrifugal forces, plasma rotation, local radiation emission to mention just a few[13].

Even if no first principle models exist, which can predict the occurrence of all the types of the plasma collapse, in the community there is a general consensus about the main stages of plasma dynamics leading to disruptions[14]. Various factors, ranging from human errors to anomalies in the radiation, temperature or density, can cause modifications of the pressure and current profiles, which destabilise macroscopic instabilities. These electromagnetic modes typically slow down and stop rotating with respect to the vacuum vessel, becoming even more unstable, to the point that the magnetic configuration is so altered that cannot be sustained. A good understanding of these physical mechanisms seems essential, to predict these catastrophic collapses and to develop real-time compatible avoidance, prevention and mitigation strategies (see caption of Fig. 1). Determining how close the plasma is to the boundary between the safe and disruptive regions of the operational space is an essential task usually referred to as proximity detection.

The subject of the present work consists of overviewing the physics of the plasma dynamics leading to disruptions, with the aim of devising reliable disruption predictors, capable of providing the

## Table 1 | Database statistics

| | C38 | C39 | C40 | C41 | Total | Baseline | Hybrid |
|---|---|---|---|---|---|---|---|
| Total | 907 | 168 | 310 | 298 | 1683 | 1324 | 359 |
| Safe | 653 | 133 | 184 | 171 | 1141 | 881 | 260 |
| Disruptive | 254 | 35 | 126 | 127 | 542 | 443 | 99 |
| Flat Top Disruption | 62 | 23 | 33 | 33 | 151 | 140 | 11 |
| Ramp Down Disruption | 192 | 12 | 93 | 94 | 391 | 303 | 88 |

Statistics of disruptions in terms of scenario and discharge phase. Only intentional disruptions and discharges missing essential measurements have been excluded. The entries cover a range of flat top currents from 1.5 to 3.8 MA and toroidal fields from 1.7 to 3.9 T. The maximum input power reached almost 40 MW. Fuel: C38 campaign in DD, C39 in DD and H, C40 in full tritium and C41 50-50 DT.

control systems of next-generation devices with enough warning time to implement realistic strategies to handle the problem. To this end, the qualitatively and quantitatively most ambitious JET database has been built and analysed with real-time compatible tools: all the discharges of JET campaigns, from the middle of 2019 to the end of 2021, have been investigated. These campaigns are particularly relevant because they cover experiments of different fuel mixtures: discharges with all the hydrogen isotopes (H/D/T) separately and the reactor 50-50 DT mixture, including the top performance experiments ever performed on JET[11]. The analysed database consists of 1683 discharges, of which 542 disruptive. The global statistics, obtained with a dedicated relational database, are reported in Table 1.

In the following, the first subsection of the Results describes the original aspects of the developed predictors. The second subsection of the Results introduces the observers developed to identify the plasma state for control purposes. In the third subsection of the Results, the modelling of the plasma dynamics leading to the main types of disruptions is discussed in detail. The investigation of the possible strategies of avoidance and prevention, which could be realistically implemented in real time in the next generation of devices, is the subject of the last subsection of the Results. The discussion section of the paper is devoted to a brief overview of the work and the lines of future research, including the prospects of transferring the present experience to future devices. More details about the diagnostics, the analysis techniques and the control tools can be found in the section 'Methods' and the supplementary material.

## Results

### An advanced, control-oriented approach to disruption prediction

The availability of reliable disruption predictors is a prerequisite to the deployment of any form of avoidance or remedial action. The development of such tools requires investigating in detail the sequence of macroscopic anomalies in the magnetic topology, kinetic profiles and radiation emission, to determine the proximity of the plasma to the disruption boundary. In the last decades, the potential of machine learning methods for this task has been studied in detail as overviewed briefly in section 'Methods' and in more detail in ref. 7. Despite quite encouraging results, disruption predictors, relying on traditional machine-learning technologies, have shown some inherent fundamental weaknesses related to: (a) the approach to learning; (b) the mathematical form of the models; (c) the choice of the input signals or features.

With regard to the first issue, most predictors, reviewed in section 'Methods', implemented a closed-world approach to learning. This method needs all the information required for the training to be available prior to the first prediction. Moreover, their performances are predicated on the assumption of absolute stationarity, in the sense that the training data, the test data and the inputs in the actual deployment must be generated by identical systems. Such a hypothesis is systematically violated, due to the rapid evolution of experimental programmes. Consequently classifiers based on closed-world training, having no capability of adapting to different regimes or unexplored physics, present some quite significant practical flaws. They tend to require very large amounts of data for the training, impossible to collect in future devices, and they lack generality, because the quality of their predictions tend to degrade quickly even when the experiments present properties only slightly different from those in the training data. Moreover, the deployment of predictors, trained in the traditional way, is typically limited to their tokamak of origin[15–17].

Another category of drawbacks is a consequence of the mathematical form of the models, implicitly assumed by the traditional machine learning tools such as SVM or neural networks. Whatever the application, most machine learning tools basically fit only a specific class of functions to the data. This means that their models have no relation to the actual plasma dynamics, are difficult to interpret and whether they can be extrapolated to future, larger devices is doubtful[18].

The third class of weaknesses is related to the choice of the input signals, which has not been always optimised. Better profile indicators are required (see section 'Methods'). Moreover, a fundamental direction for future improvements involves a better detection of radiation anomalies, which are localised patterns of high emission. Most disruptions are indeed preceded or caused by anomalous radiation events, which often constitute the earliest macroscopic precursors by a quite ample margin[19]. The three most common anomalous radiation patterns leading to disruption on JET are reported in Fig. 2 (and the tools developed to detect them in real time are described in detail in section 'Methods').

The consequences of the just described limitations are particularly unsatisfactory also because natural forms of intelligence can cope quite well with these cognitive problems. Various animals, and humans in particular, can learn effectively from few examples, can adapt quickly to changing situations and have also the capability of transferring knowledge from one problem to similar ones. It is the contention of the present work that these gaps between natural and artificial forms of intelligence can be reduced, by developing adaptive tools, less abstract and closer to the actual physics and dynamics of the phenomena under study. They will require original data processing techniques to analyse and interpret the available measurements in real time. In this perspective, the tools developed in the context of the present work rely on real time compatible measurements and signal processing techniques and present the following four innovative characteristics: (1) they utilise physics-based but control-oriented indicators as inputs (see section 'Methods'); (2) they output not only alarms but also a classification of the type of anomaly detected (see section 'Methods' for details); (3) they implement an automated adaptive form of training to follow the evolution of the experimental programme see (section 'Methods'); (4) they operate in the two-dimensional space of probability and time; they therefore provide both the disruption probability and an estimate of the time remaining before the beginning of the current quench inputs (see section 'Methods' for the details). Of course, all these aspects are of great help for the control system to optimise any form of remedial strategy.

### Characterising the plasma state for proximity control

The research in the last years has revealed that the essential diagnostics, for proximity control and mitigation, are: global indicators of the magnetic configuration state, the kinetic profiles, visible imaging and bolometric tomography. Their role in predicting the occurrence of disruptions is overviewed in this subsection.

Global indicators of the magnetic configuration are important to determine the proximity of the plasma to the disruption boundary (see section 'Methods'). Two quantities, which have proved very useful, and which will be available also in the next generation of devices, are the normalised internal inductance $l_i$ and the safety factor at 95% of the plasma minor radius $q_{95}$. The best estimates of $l_i$ and $q_{95}$ are the output of quite sophisticated equilibrium codes, whose description is beyond the scope of the present work[5]. However, intuitively the internal inductance provides an indication about how peaked the current profile is. Small values of $l_i$ correspond to a broad current profile. This is important for stability and to interpret other signals such as the locked mode amplitude (see later and ref. 20). The safety factor indicates how much stabilising toroidal field is used in a certain configuration, again quantifying the vulnerability of the configuration to become instable. Other things being equal, plasmas at higher $q_{95}$ are more stable. At values of $q_{95} \approx 3$, the reference of ITER baseline

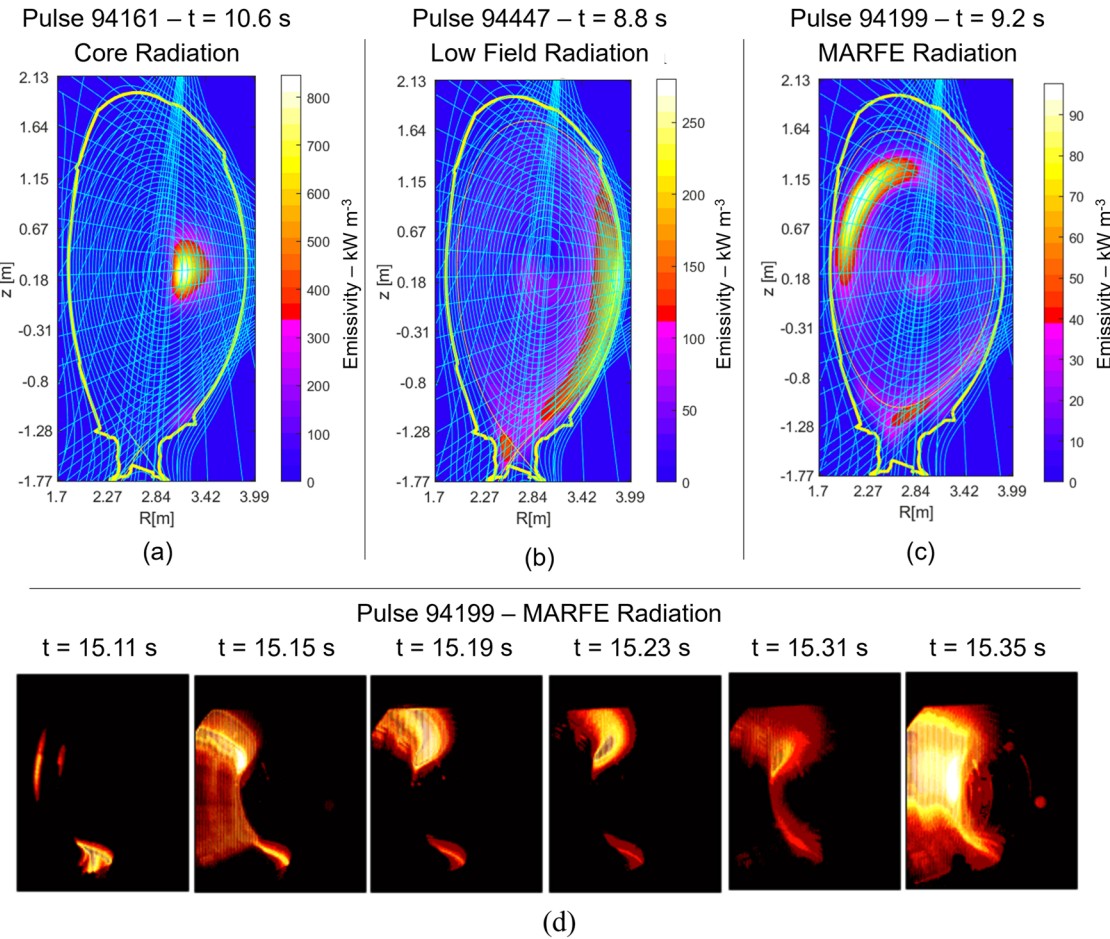

**Fig. 2 | Main radiation patterns.** The main radiation patterns leading to a radiation collapse of the plasma in JET with a metallic wall, as revealed by the maximum likelihood tomography and visible imaging. **a** Anomalous high radiation in the core due to accumulation of high Z impurities, mainly W. **b** Anomalous low field radiation. The crescent shape excessive emission of is typically an effect of heavy impurities in configurations, in which, due to the combination of neoclassical and anomalous transport, they accumulate in the outer equatorial plane. **c** Radiation instability of the MARFE (Multifaceted Asymmetric Radiation from the Edge) type[14,15], typically consequence of impurity seeding or excessive high density in the attempt to reduce the power load on the divertor. **d** MARFE radiation observed by the visible camera (colour proportional to pixel intensity).

scenario, the plasma disruptivity of present-day tokamak can exceed 60%, absolutely incompatible with the operation of a reactor level device. When macroscopic electromagnetic instabilities, capable of leading to disruptions, stop rotating in the frame of reference of the lab, their amplitude increases and their magnetic signature is picked up by large saddle coils located around the vacuum vessel. This so-called Locked Mode (LM) amplitude signal is important because almost all disruptions on JET are preceded by an anomalous increase of this quantity[21].

Anomalies in the electron temperature profile can also play a fundamental role in destabilising various macroscopic instabilities, namely MHD modes, which then grow, lock to the wall and finally lead to disruptions[14]. The so called *electron temperature hollowness* and *edge cooling* are the two most evident and potentially harmful temperature anomalies.

The hollowing of the electron temperature $T_e$ is a situation, in which the maximum temperature is not reached in the centre of the plasma but off-axis, resulting in a double bumped profile (see section 'Methods'). In metallic devices, such an unhealthy plasma condition is typically caused by an excessive emission of radiation in the core, mostly a consequence of the accumulation of heavy impurities, mainly tungsten. The consequent modifications of the local resistivity $\eta \sim Z_{eff}/T_e^{3/2}$, cause changes of the plasma current profile, which often lead to the destabilisation of macroscopic modes[14].

Analogous destabilising effects, due to the alterations of the current profile, may be caused by the dynamics at the plasma edge. In this case, excessive local density or radiation, typically due to the attempt to reduce the power loads on the divertor, induce a reduction in the peripheral temperature, with consequent contraction of the current profile, leading to tearing mode destabilisation[14].

An important point to appreciate is that unhealthy modifications of the temperature profile can occur significantly in advance of the destabilisation of the fatal MHD modes, giving more time to intervene. The availability of robust and reliable indicators of anomalies in the electron temperature profile is therefore a crucial ingredient in the development of predictors for prevention and avoidance. The ones implemented in the present work are overviewed in the section 'Methods' and in ref. 22.

In the whole analysed database (see introductory section), only 7 disruptions do not show any anomalous radiation pattern. Properly measuring the total emission of radiation will be therefore a crucial and delicate task in all future machines. The main radiation patterns leading to the plasma collapse on JET are reported in Fig. 2. All these emission events are cases of radiation positive feedback instabilities, because the impurities have a radiation function increasing with the inverse of the temperature. Consequently, when the local radiation is sufficiently high to provoke a reduction in the temperature, the impurities radiate more, causing a further

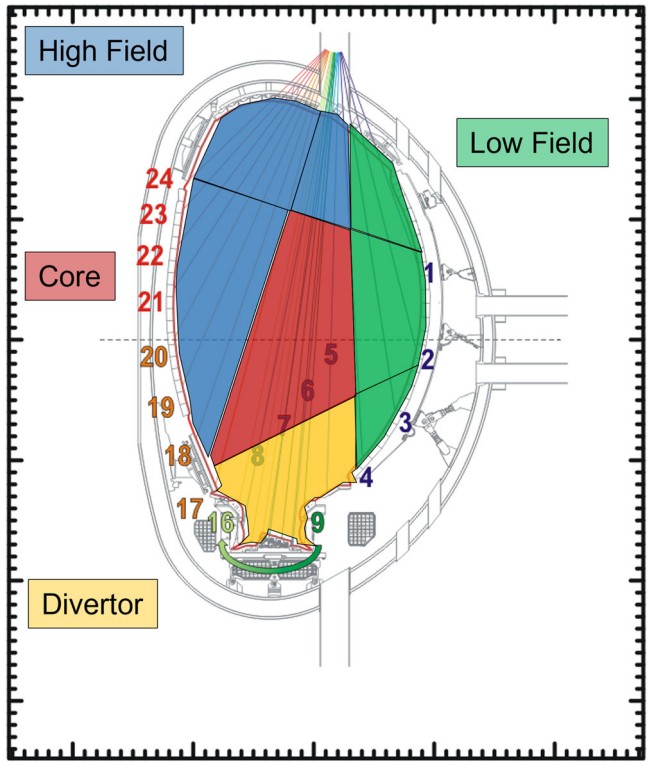

**Fig. 3 | Fast time resolution tomography regions.** The four macro regions, whose emitted power is determined with the fast time resolution tomography described in the section 'Methods'.

decrease of the temperature, which results in higher emission, in a vicious positive feedback loop. The most important point to appreciate is that the radiation anomalies are normally the first ones to appear and therefore it is essential to detect them for avoidance and prevention purposes (see next subsection).

The accurate determination of the radiation losses on JET requires tomographic inversion of the bolometric signals, which is a very ill-posed problem. This aspect has been addressed on JET with the combination of a sophisticated tomographic method, called the Maximum Likelihood (ML), and a fast high time resolution algorithm for deployment in real time (see section 'Methods'). Most radiation problems in the high-field side (such as Multifaceted Asymmetric Radiation from the Edge or MARFEs) appear also very clearly in the frames of the visible cameras and do not require particularly sophisticated image processing algorithms to be followed in real time (see section 'Methods').

**Modelling the plasma thermal dynamics leading to disruptions**
To predict and interpret the plasma evolution in the precursor phase leading to the disruption, a model of the radiation dynamics is necessary. The one described in the present subsection has been conceived with real-time applications in mind; attention has been paid to finding an appropriate trade-off between accuracy and real-time compatibility. In this perspective, to model quantitatively the processes in Fig. 2 for prediction purposes, the most suitable indicator proves to be the ratio of the radiated power divided by the plasma thermal energy. This quantity is much more useful than the traditional ratio of radiated power divided by input power, which can diverge and become misleading, for example during the ramp down of the plasma current, when the additional heating systems are switched off [23]. In terms of this quantity, the balance equation, to analyse the radiation anomalies leading to radiative collapse, can be written as:

$$\frac{1}{E_{\mathrm{p}}(\rho,\theta)}\frac{dE_{\mathrm{p}}(\rho,\theta)}{dt} = \frac{P_{\mathrm{in}}(\rho,\theta)}{E_{\mathrm{p}}(\rho,\theta)} - \frac{P_{\mathrm{rad}}(\rho,\theta)}{E_{\mathrm{p}}(\rho,\theta)} + \frac{P_{\mathrm{t}}(\rho,\theta)}{E_{\mathrm{p}}(\rho,\theta)} \tag{1}$$

Where $E_{\mathrm{p}}$ is the plasma thermal energy, $P_{\mathrm{in}}$ the input power, $P_{\mathrm{rad}}$ the local radiated power, $P_t$ the power due to transport. Equation (1) is particularised for each macropixel of the plasma cross-section and $\rho$ and $\theta$ are the polar coordinates of the considered region barycentre (see Fig. 3). The shape and size of the macro-pixels are determined by the layout of JET bolometric diagnostic (see section 'Methods'). The local plasma energy has been calculated employing the measurements of the High Resolution Thomson Scattering (HRTS) over the macro-pixels, under the usual approximations that electrons and ions have the same temperature and density i.e. $T_e = T_i$, $n_e = n_i$ (see section 'Methods').

Within the help of Eq. (1) and the high time resolution tomography, it has been possible to study the dynamics of the plasma leading to the disruptions in hundreds of discharges. A threshold level $\Lambda_{\mathrm{ith}}$ of $\Lambda_i=P_{\mathrm{rad}}/E_{\mathrm{p}}$, for each of the 4 regions shown in Fig. 3, can be set by the user, depending on the needs of the experimental programme and its potential danger for the device. In the following, the reported threshold levels $\Lambda_{\mathrm{ith}}$ have been chosen so that, above the corresponding value of the ratio, the plasma has more than 50% probability to be in a state with excessive radiation, tending to develop temperature and magnetic anomalies. The details about the calculation of the various disruption probabilities are provided in section 'Methods'.

The distribution of the first alarms, due to radiation anomalies in disruptive discharges, is reported in the top histogram of Fig. 4. It has been checked on a well-documented set of discharges that the alarms of the predictors developed in the present work precede the interventions of JET control system in practically all cases, as documented in section 'Methods'. In all regions, significantly more disruptions occur in the ramp down of the plasma current than during the flat top. Radiation anomalies appear also in discharges that in the end do not disrupt, as reported in the bottom histogram of Fig. 4. In these occurrences, the safe ending of the pulse is often due either to positive interventions of the control system or to rapid terminations of the discharge, not allowing the plasma dynamics to evolve naturally. Moreover, it has also been verified that remedial actions, of the type advocated in the next subsection, would be always beneficial and improve the performance of the plasma even if implemented in the discharges not disrupting. Indeed, after the appearance of the detected radiation anomalies, if not remedial actions of the type proposed in this work were taken (see next subsection), these experiments would be typically compromised anyway, even if they did not disrupt, and would not provide any useful information.

Since $1/\Lambda_i$ has the dimension of time, it can be interpreted as a cooling time. As mentioned, the traditional approaches to proximity assessment estimate either the probability of disruption or the time remaining to the onset of a macroscopic MHD mode. With the model just described it is possible to combine these two quantities and operate in the bi-dimensional space of time and probability.

It should be emphasised that all the results reported in the present work have been obtained with a fully adaptive technique from scratch to update the $\Lambda_i$ values, as described in section 'Methods'. The first guess has been obtained with only the first disruptive and three non-disruptive discharges, and the thresholds have been updated with adaptive techniques on a shot-to-shot basis [15–18]. It is also important to mention that all the proposed indicators and used measurements are available in real time. All the quantities utilised in this paper can be comfortably calculated within JET real time network cycle time of 2 ms.

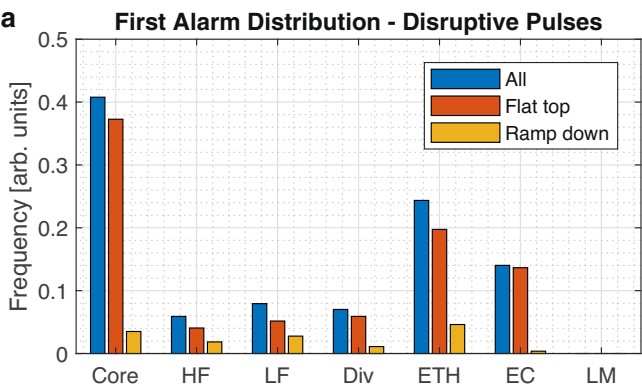

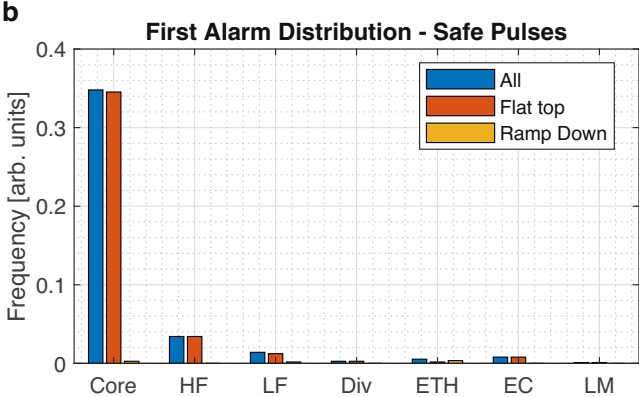

**Fig. 4 | Alarm statistics. a** Distribution of the first alarms in disruptive pulses, particularised for the current flat top and ramp down. **b** Distribution of the first alarms in safe pulses, also particularised for the current flat top and ramp down. The locked mode does not trigger any false alarms, therefore not causing any unnecessary mitigation. The majority of the first alarms in safe discharges are due to the appearance of blobs of radiation in the core. The presence of such anomalies has been checked with the ML tomography; consequently these alarms would not result in negative interventions by the control system, because the proposed remedial action of increasing the power deposition in the core would only improve the plasma performances.

## Table 2 | Sequence of anomalies

| | |
|---|---|
| Hollowness preceded by Core Radiation | 52.9% |
| Hollowness preceded by LF Radiation | 65.2% |
| Hollowness preceded by Core or LF Radiation | 90.5% |
| Edge Cooling preceded by HF Radiation | 64.4% |
| Edge Cooling preceded by LF Radiation | 51.3% |
| Edge Cooling preceded by Core or LF Radiation | 79.2% |
| Mode Locking preceded by Hollowness | 36.0% |
| Mode Locking preceded by Edge Cooling | 40.6% |
| Mode Locking preceded by Radiation Anomalies | 81.7% |

Sequence of the anomalies with percentages. The locked mode alarms are practically always preceded by other anomalies.

## Table 3 | Time between anomalies

| | Mean (ms) | Std (ms) |
|---|---|---|
| $\Delta t_{\Lambda,core} - \Delta t_{ETH}$ (all) | 167 | 270 |
| $\Delta t_{\Lambda,core} - \Delta t_{ETH}$ (FT) | 139 | 435 |
| $\Delta t_{\Lambda,core} - \Delta t_{ETH}$ (RD) | 176 | 190 |
| $\Delta t_{\Lambda,HF} - \Delta t_{EC}$ (all) | 208 | 467 |
| $\Delta t_{\Lambda,HF} - \Delta t_{EC}$ (FT) | 206 | 522 |
| $\Delta t_{\Lambda,HF} - \Delta t_{EC}$ (RD) | 208 | 454 |
| $\Delta t_{\Lambda,LF} - \Delta t_{ETH}$ (all) | 568 | 778 |
| $\Delta t_{\Lambda,LF} - \Delta t_{ETH}$ (FT) | 306 | 472 |
| $\Delta t_{\Lambda,LF} - \Delta t_{ETH}$ (RD) | 690 | 859 |
| $\Delta t_{\Lambda,LF} - \Delta t_{EC}$ (all) | 700 | 933 |
| $\Delta t_{\Lambda,LF} - \Delta t_{EC}$ (FT) | 300 | 588 |
| $\Delta t_{\Lambda,LF} - \Delta t_{EC}$ (RD) | 784 | 972 |
| $\Delta t_{\Lambda,ETH} - \Delta t_{LM}$ (all) | 1069 | 1003 |
| $\Delta t_{\Lambda,ETH} - \Delta t_{LM}$ (FT) | 1105 | 1131 |
| $\Delta t_{\Lambda,ETH} - \Delta t_{LM}$ (RD) | 1062 | 983 |
| $\Delta t_{\Lambda,EC} - \Delta t_{LM}$ (all) | 259 | 659 |
| $\Delta t_{\Lambda,EC} - \Delta t_{LM}$ (FT) | 304 | 716 |
| $\Delta t_{\Lambda,EC} - \Delta t_{LM}$ (RD) | 236 | 631 |
| $\Delta t_{\Lambda,ALL} - \Delta t_{LM}$ (all) | 773 | 952 |
| $\Delta t_{\Lambda,ALL} - \Delta t_{LM}$ (FT) | 604 | 857 |
| $\Delta t_{\Lambda,ALL} - \Delta t_{LM}$ (RD) | 838 | 981 |

The time intervals between the various anomalies: FT indicates the flat top and RD the ramp down phase of the discharges. Since the time intervals are never negative, large standard deviations are to be considered positive, because they indicate that in many cases the control system would have plenty of time to intervene.

## The strategy for proximity control

Coming to the relationship between the radiation emission and the dynamics of the plasma kinetic quantities, the statistical results, covering the sequence of the various anomalies, are reported in Table 2. The intervals between the various types of anomalies are overviewed in Table 3.

The collected evidence indicates that the increase in the locked mode is almost always preceded by some other anomalies. Excessive radiation alarms precede the temperature hollowness in about 90% of the cases. Radiation anomalies in the edge are detected earlier than edge cooling in about 80% of occurrences. The intervals between the preceding alarms and the subsequent ones vary but are typically enough to undertake remedial actions (before having to trigger mitigation).

Coming to the exploitation of the evidence just discussed for control purposes, the approach pursued in the present work consists of estimating the proximity to the disruption boundary in three different steps. First, the radiation patterns and their criticalities are estimated, secondly, the anomalies of the temperature profiles are assessed. The healthiness of magnetic configuration is determined last.

With these objectives and in the light of the evidence just overviewed, a reasonable control strategy can be devised, whose main elements are shown in the block diagram of Fig. 5. Starting from the beginning of the current quench and moving backward in time, the

approach could consist of triggering mitigation actions immediately after a locked mode alarm. Following a warning due to excessive cooling of the edge, the time remaining before the beginning of the current quench is compatible only with prevention and therefore the control system should immediately activate the sequence of actions to terminate the discharge safely. In case of anomalies in the radiation patterns, the heating is to be increased in the affected regions. With sufficiently flexible additional heating schemes, such as those already deployed to avoid MHD instabilities[24,25], the regions of excessive radiation could be targeted, avoiding the radiation collapse (and consequently avoiding the disruption).

The proposed control logic has been implemented in fully compatible real-time conditions, which means that all diagnostics, indicators and predictors would have worked exactly in the same way in closed feedback loop. Assuming that 100 ms warnings for temperature hollowness and edge cooling are sufficient to undertake successful remedial action (and 10 ms warning time is enough for mitigation), for

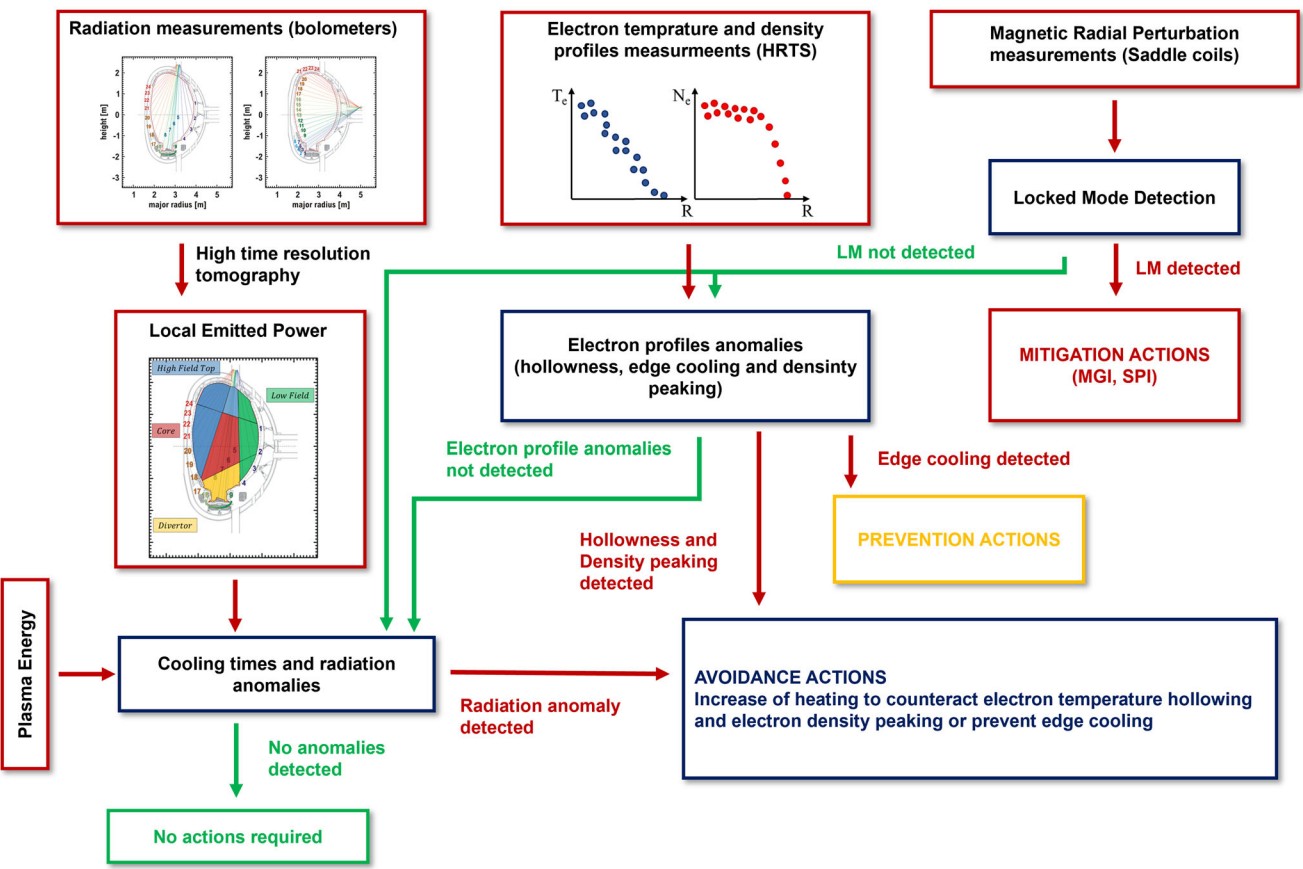

**Fig. 5 | Control logic.** Block diagram of the detection and control logic proposed in the present work.

the considered database the proposed control logic provides the performances reported in Tables 4 and 5. Keeping in mind that the predictors implementing the technologies developed in refs. 15,21 are almost 100% successful for mitigation, the outcomes of the proposed strategy look quite promising. Indeed more than 50% of the disruptions could be avoided and a quarter prevented. It should also be emphasised that, in almost all the safe discharges, in which unnecessary mitigations would have been triggered, the alarms are raised after the intervention of JET control system and therefore these plasmas do not disrupt only because termination actions had already been undertaken. All things considered, therefore the obtained results would be compatible with ITER requirements. It is also worth noting that about 85% of the first warnings, triggered by the proposed detection strategy, precede any action of JET control systems (the Real Time Central Controller RTCC and the Plasma Event TRiggering of Alarms PETRA[26,27]) and are therefore reaction to the natural evolution of the plasma (a detailed comparison of the proposed tools' performances and the interventions of JET control system is the subject of section 'Methods'). Some examples of JET discharges, proving the viability of the proposed strategy, are reported in section 'Methods' as well.

It should also be mentioned that, by choosing different probability thresholds, the user can find a different trade-off between the need for mitigation and the risk of perturbing discharges that would not disrupt. The estimates of the time to the onset of a MHD macroscopic instabilities could also allow further optimisation of the control strategies. The quality of such estimates, obtained with a dedicated neural network, are described in section 'Methods', from which it can be concluded that the accurate measurement of the radiation losses could help significantly in deciding the best actions to take particularly for avoidance. It is worth mentioning that information about the time

to the disruption has not been used in the logic of the controller because this is an aspect, whose transfer from one device to another has not been proven yet[18].

## Discussion

Understanding and predicting the sequence of macroscopic anomalies leading to disruptions will be essential elements of any realistic strategies of proximity control and avoidance in the next generation tokamaks. The presented work shows that, to achieve these tasks, sufficiently resolved measurements of the magnetic configuration, kinetic quantities and radiation emission are essential. The derived indicators have to be physics based but at the same time sufficiently robust and informative about the future evolution of the plasma, to constitute good inputs to the control system. The probability of disruption allows optimising the remedial strategies and fine tuning them depending on the specific needs of the experiments. The estimates of the time remaining to the beginning of the current quench constitute another important piece of information for optimising the control system interventions.

From a diagnostic perspective, the spatially involved nature of the various paths to collapse implies that the tokamak, even if not spatially complex as the stellarator, requires at least 2D measurements to be fully understood and controlled, not an easy proposition for the reactor. However, better diagnostic capability of the emitted radiation would provide essential advantages in terms of disruption prevention and avoidance. Bolometry and visible imaging, if implemented with the appropriate time and spatial resolution, can indeed provide significantly earlier warning that the discharge is drifting toward dangerous operational conditions. Careful detection of the kinetic profiles main features will also be essential to optimise the remedial strategies. Indeed in the reactor false alarms will have very undesirable

**Table 4 | Warning time statistics**

| | Mean [ms] | Std [ms] |
|---|---|---|
| $\Delta t_{\Lambda,core}$ (all) | 676 | 807 |
| $\Delta t_{\Lambda,core}$ (FT) | 240 | 531 |
| $\Delta t_{\Lambda,core}$ (RD) | 825 | 832 |
| $\Delta t_{\Lambda,HF}$ (all) | 398 | 564 |
| $\Delta t_{\Lambda,HF}$ (FT) | 216 | 372 |
| $\Delta t_{\Lambda,HF}$ (RD) | 471 | 610 |
| $\Delta t_{\Lambda,LF}$ (all) | 699 | 919 |
| $\Delta t_{\Lambda,LF}$ (FT) | 326 | 545 |
| $\Delta t_{\Lambda,LF}$ (RD) | 852 | 995 |
| $\Delta t_{\Lambda,Div}$ (all) | 492 | 755 |
| $\Delta t_{\Lambda,Div}$ (FT) | 378 | 739 |
| $\Delta t_{\Lambda,Div}$ (RD) | 539 | 757 |
| $\Delta t_{ETH}$ (all) | 936 | 1066 |
| $\Delta t_{ETH}$ (FT) | 545 | 966 |
| $\Delta t_{ETH}$ (RD) | 1060 | 1069 |
| $\Delta t_{EC}$ (all) | 390 | 590 |
| $\Delta t_{EC}$ (FT) | 545 | 988 |
| $\Delta t_{EC}$ (RD) | 334 | 341 |
| $\Delta t_{LM}$ (all) | 282 | 402 |
| $\Delta t_{LM}$ (FT) | 211 | 442 |
| $\Delta t_{LM}$ (RD) | 309 | 382 |

The time between the detection of the various anomalies and the beginning of the current quench. FT indicates the flat top and RD the ramp down phase of the discharges.

**Table 5 | Control performances**

| | All | FT | RD |
|---|---|---|---|
| Disruptive | 542 | 27.9% | 72.1% |
| Mitigated | 35.1% | 13.5% | 21.6% |
| Prevented | 14.9% | 3.8% | 11.1% |
| Avoided | 49.3% | 9.8% | 39.5% |
| Tardy | 0.7% | 0.7% | 0.0% |
| Missed | 0.0% | 0.0% | 0.0% |
| Safe | 1141 | – | – |
| Mitigated | 9.0% | 1.3% | 7.7% |
| Prevented | 7.9% | 1.6% | 6.3% |
| Avoided | 18.3% | 18.0% | 0.3% |
| No Alarm | 64.8% | – | – |

Performances of the developed control strategy for the entire database summarised in Table I. The table reports the percentages of the various interventions that would have been achieved with the implemented values of the hyper-parameters: 100 ms minimum time to perform avoidance and prevention actions and 10 ms minimum warning time for successful mitigation.

consequences as well, because they might interrupt the production of electricity in critical moments.

It is also worth emphasising that the indicators devised do not show any significant dependence from the isotopic composition. This is a very important aspect, because it can potentially lead to feedback strategies, which are not too dependent on a plasma quantity, the fuel mixture, which is very delicate to control. Since the tools developed in this work are all adaptive and real-time compatible, they constitute a quite encouraging package. Indeed, they have managed to follow the evolution of the experimental programme in probably the most ambitious and varied set of campaigns ever carried out on JET. Moreover, given the fact that the indicators tested are based on normalised quantities or ratios, they are expected to be easily transferrable to other devices. Indeed, the implementation of

some of the proposed tools on ASDEX Upgrade (the second largest metallic device in Europe), the second largest metallic tokamak in Europe, has already started and the preliminary indications are very positive.

The proposed control logic, implemented in completely real-time compatible conditions, results in no missed alarms, 65% of avoided and prevented disruptions and almost no wrong actions. Consequently, the developed tools, which work equally well with all reactor-relevant hydrogen fuel mixtures, are expected to satisfy ITER requirements, which will have better actuators to implement any remedial action. On the other hand, in the future, the technique will have to be tested and optimised in more reactor-relevant conditions, i.e. in experiments at high density[28,29], with full detached divertors and radiated fraction of about 90%. Small ELMs regimes and negative triangularity configurations will require specific and detailed investigations as well.

From a methodological perspective, a potentially very significant improvement would consist of state aware tools, particularised for the main phases of the discharge: ramp up, flat top and ramp down of the plasma current. Indeed the plasma behaviour in these different phases is not exactly the same and the performance of specific predictors could be significantly better than those of the general ones developed so far. Such a step would require implementing trajectory learning solutions, i.e. techniques that can take into account the history of the discharge and that do not consider only individual time slices of the data[15,16,30,31].

## Methods
### Brief history of machine learning-based disruption predictors in tokamaks

Reliable prediction is an essential ingredient of any strategy to overcome the problem posed by disruptions. Unfortunately, there are no first principle plasma models capable of predicting the occurrence of disruptions even offline, let alone in the short times required for real time applications. The international community, recognising the importance of developing reliable predictors, has promoted the development of data-driven empirical models since some time. The simple ones, devised manually, unfortunately have provided very disappointing performances particularly on JET with a metallic wall[20]. Consequently, given also the huge amounts of data collected in modern devices, data mining techniques have become increasingly popular in the last decades. These methodologies will become even more crucial in the future since, for example, ITER is expected to produce 50 Gbytes of data per second (2 Petabytes per day), an amount of information impossible to process with traditional, human-centred techniques[32].

Disruption prediction is usually conceptualised as a binary classification problem. The machine learning tools are required to divide the operational space into two regions, the disruptive and the safe one. Various forms of training have been tested, from traditional supervised to adaptive, including even reinforcement learning strategies[33]. The derived models of the boundary are then to be deployed in real time during the discharges with a typical time resolution of the order of milliseconds.

The signals to be processed are time series, consisting of sequences of data points indexed in time. The choice of the features to be provided as inputs to the predictors are typically identified manually by the experts and require various forms of pre-processing. All the main real-time signal processing techniques available both in the time domain[34–44] and in the frequency domain[45] have been implemented. Approaches based on mixture of time and frequency domains, such as wavelet transforms, have not been neglected either[46].

So far, all the main families of existing real-time compatible classifier technologies have been explored, including support vector

machines, artificial neural networks, generative topographic mapping, fuzzy logic and deep learning. They have also been applied to the data of many tokamaks of various generations: ADITYA (India)[47], ASDEX Upgrade (Germany)[48], DIII-D (US)[49–51], J-TEXT (China)[52], NSTX (US)[53], ALCATOR C-MOD (US)[54], JT-60U (Japan)[55], EAST (China)[56–58], HL-2A (China)[59] and JET (UK)[7]. Up to now only three machine-learning-based predictors have been implemented in JET's real-time network, APODIS[60], SPAD[61], and the centroid-based method[21].

### The requirements of diagnostics and data analysis tools

In general, the diagnostics for disruption prediction have to provide more stringent performances than the measurements for the understanding of the physics. Certainly the RAMI (Reliability, Availability, Maintainability, Inspectability) risk control requirements are much more severe, because the needed measurements have to be always available in practically all phases of the experiments[62]. Spatial and time resolutions have also to be carefully considered, since the causes of disruptions are multiple, some quite local and evolving on fast time scales. On top of that, depending on their objectives (mitigation, prevention or avoidance), the various predictors have to rely on different signals. In particular, the measurements for proximity control, for assessing how close to the disruption boundary the plasma state is, have to provide sufficient warning times. For mitigation, the premium of course is on accuracy, since non-mitigated disruptions are particularly bad and probably cannot be tolerated in devices of the size of DEMO[12].

Whatever the source of information, the adopted metrics to analyse the experimental signals for disruption prediction should present the following desirable properties:

1. Sensitive, in the sense of being capable of detecting the anomalies at an early stage.
2. Specific, which means triggering a very small number of false detections.
3. Deployable in real-time.
4. Easily portable from one tokamak to another.

All the diagnostics and derived quantities proposed in the present work satisfy these criteria quite well, as discussed in detail in the following subsections. They are also expected to be available in DEMO and the reactor. The order follows the time proximity to the beginning of the current quench: mitigation, prevention and then avoidance.

### Diagnostics of the magnetic configuration.

Following electrical engineering standards, the internal inductance is based on the magnetic energy W stored in the poloidal field in the region enclosed by the plasma boundary. Its normalised version, used for ITER design, is defined as $l_i = \frac{2V<B_p^2>}{\mu_0^2 I^2 R_0}$ where $V$ is the plasma volume, $R_0$ is the major radius of the magnetic axis, $B_p$ the poloidal component of the magnetic field, <> indicate volume average, $I$ is the plasma current and $\mu_0$ is the vacuum magnetic permeability.

The safety factor essentially measures the "windiness" of the magnetic field at a certain radius and in general can be conceived as its pitch: $q = d\phi/d\theta$ where $\phi$ is the toroidal angle and $\theta$ the poloidal one (see Fig. 1a). In practice it is typically expressed in terms of the fields as $q = rB_\phi/RB_\theta$ where r is the minor radius, $R$ the major radius, $B_\phi$ and $B_\theta$ are the toroidal and poloidal component of the magnetic field respectively. The indicator $q_{95}$ is the value of the safety factor at 95% of the plasma radius, which has proved to be a very good indicator of the plasma stability.

As mentioned, the locked mode amplitude is the most important signal for mitigation on JET[7]. The relationship between $l_i$ and the locked mode was investigated in ref. 20. Recently the slowing down of the large electromagnetic modes, preceding the disruption, has been detected using this signal[63]. On the other hand, the anomaly in the

locked mode amplitude typically manifests itself too late for prevention and avoidance and therefore this signal is useful only for mitigation. Indeed, macroscopic alterations to the current profile and the destabilisation of macroscopic electromagnetic instabilities are typically the last stage of the pre-disruptive plasma evolution.

### Diagnostics of the kinetic profiles.

A literature survey has been performed to investigate the performances of the available kinetic profile indicators for both anomalous behaviours: the hollowing of the profile in the core and the cooling of the edge. With regard to the core hollowness, all the most widely used criteria rely either on the ratios of the temperatures in different regions of the plasma (the "core vs all" peaking factor[64] or the core-middle-edge average temperature ratios[14,65]), or on the combination of cumulative profiles obtained weighting differently the core and the edge[66] An innovative indicator, called Gaussian Fitted Hollowness (GFH), has been recently devised, in the framework of the present work[22]. The approach consists of fitting the electron temperature profile with a bimodal Gaussian. Indeed, healthy, safe electron temperature profiles are unimodal, i.e. present a peak only at or near the plasma centre, whereas hollow profiles are bimodal (see Supplementary Fig. 1). Consequently, detecting and quantifying bimodality is an alternative way to determine the degree of hollowness. The developed indicator relies on the Bhattacharya distance ($D_B$)[67] which was originally devised to quantify the distance between two Gaussian distributions and is defined as:

$$D_B = \frac{1}{4}\log\left(\frac{1}{4}\left(\frac{\sigma_1^2}{\sigma_2^2} + \frac{\sigma_2^2}{\sigma_1^2} + 2\right)\right) + \frac{1}{4}\left(\frac{(\mu_1 - \mu_2)^2}{\sigma_1^2 + \sigma_2^2}\right) \quad (2)$$

Where $\mu_i$ are the centres and $\sigma_i$ the standard deviations of the Gaussians.

Given the symmetries imposed by the topology of present day and future devices, it has proved sufficient to fit the profile with a symmetric bimodal Gaussian function ($A_1 = A_2 = A, \mu_1 = -\mu_2, \sigma_1 = \sigma_2$). Particularised for this case, the Bhattacharyya distance becomes $D_B = \frac{\mu^2}{2\sigma^2}$ and therefore the GFH has been defined as:

$$GFH = \frac{\mu}{\sigma} = \sqrt{2D_B} \quad (3)$$

The GFH indicator is therefore the distance, between the peak and the valley of the bimodal distribution, divided by the standard deviation of the single Gaussians (see Supplementary Fig. 1). A comparison of the performance of the proposed indicator and the most widely ones used in the community is reported in the Supplementary Fig. 2. These results have been obtained with a series of numerical tests and confirmed by the analysis of a comprehensive set of JET discharges[22]. The superiority of the proposed GFH indicator emerges quite clearly from all the relevant metrics considered.

Criteria for the edge cooling are much less developed than those for the hollowness. In the literature[12] it is reported that on JET the edge cooling indicator traditionally used relies on the average temperature in the middle region divided by the average temperature in the external region. Since its performances are not satisfactory, a series of alternatives has been explored. The most performing indicator found computes the cumulative distribution function (CDF), normalised to its maximum. On the basis of the CDF, it is possible to calculate the plasma radius ($\rho_{98}$) at which the CDF reaches the value of 98%:

$$CDF_T(r) = \int_{-a}^{r} T_{e,norm} dr \rightarrow$$
$$\rightarrow \int_{-a}^{\rho_{98}} T_{e,norm} dr = 0.98 \quad (4)$$

Since this radius shrinks with the plasma cooling down, the Cumulative Based Cooling (CBC) parameter $CBC = 1/\rho_{98}$ is calculated, to obtain a quantity increasing with the edge cooling for coherence

with the literature. The performances of the proposed indicator are compared with the most widely used alternatives in Supplementary Table 1. Again this index outperforms quite systematically all the indicators reported in the literature[22].

The density profile is much more resilient to perturbations than the temperature one. Incipient anomalies in the density at the edge are typically detected much earlier and more reliably by measurements of the radiation (for example MARFEs, see next subsection). The core density also tends to react more slowly than the temperature and becomes hollow less frequently. Density profile indicators are therefore less useful for both proximity control and disruption prediction. However, to preserve the pressure, the hollowing of the temperature is often accompanied by a peaking of the density profile. A reliable density peaking factor can therefore be a useful complement to the temperature profile indicators just described. In this perspective, given the basic consistency of the density profiles, a ratio based on the average temperature within 0.3 and within 0.8 of the normalised radius has proved to be more than adequate:

$$N_{peaking}(t) = \frac{\frac{1}{0.3}\int_0^{0.3} n_e(r,t)dr}{\frac{1}{0.8}\int_0^{0.8} n_e(r,t)dr} - 1 \tag{5}$$

With regard to the computational times, by far the most demanding indicator is GHF. Even this one though requires of the order of 1 ms to be calculated on a lap top computer with a Matlab routine. Consequently, all the developed profile indexes are fully compatible with any realistic real-time application.

**Diagnostics of the radiation emission.** The next generation of devices will have to operate at radiated fractions of about 90% and detached divertors. Routinely available estimates of the radiation emission and its accuracy will therefore be essential to control these plasmas and ensure safe operation. At the edge, early detection of MARFEs onset is particularly relevant. The main idea behind the proposed method relies on the already mentioned observation that MARFE instabilities typically start developing in the divertor region or just above it (X-point radiator)[68]. In a safe shot, the centroid of the visible radiation would remain close to the divertor throughout the entire shot. On the contrary, in disruptive discharges the blob of high radiation typically moves up and down along the wall on the high field side. Consequently, to detect the onset of a MARFE, the coordinates of the emission centroid are evaluated and compared with the ones of the previously processed frame. The developed algorithm acts on individual frames and converts them to a grey-scale image. After thresholding, the resulting binary image contains the brightest zones of the original frame. On JET, when the centroid of the high emitting region moves vertically more than 20 pixels between two frames (corresponding to about 15 cm in physical space), a MARFE instability has typically started to develop[68].

A database of JET videos has been built, to evaluate the performances of the proposed MARFE detection algorithm. A total of 44 shots, 28 disruptive and 16 safe, have been carefully selected from the high power deuterium, hydrogen and full tritium campaigns (C39 and C40). The discharges have been chosen to cover a wide range of experimental conditions, to prove the general applicability of the algorithm. For these two campaigns, videos from the wide angle operational visible camera have been analysed. For each video, the presence or absence of a MARFE has been evaluated by an expert. With the aforementioned threshold, the algorithm manages to identify all the MARFEs in the database correctly, without any false positive or false negative[68]. Also in this case, the processing times are of the order of one ms by MATLAB routines, therefore much shorter than the camera frame rate and not problematic to implement in real time.

Even if the visible cameras can allow following quite well the movements of MARFEs, the determination of the actual power emitted

requires bolometric measurements. For the purposes of control, an estimate of the uncertainties in the reconstructed emissivity is also extremely relevant. The only tomographic inversion method that can provide this information routinely is based on the Maximum Likelihood (ML). The implementation of the ML algorithm with computerized tomography is well-documented in the literature[69]. The technique was originally based on the assumption that both each of the "n" pixel of the emissivity $f_n$ and the line integrated measurements $g_m$ follow poissonian statistics. It has been demonstrated that this hypothesis can be relaxed and the method can be extended to measurements obeying Gaussian distributions[70]. The conditional probability of detecting **g** measurements from a two-dimensional emissivity **f**, can therefore be written as the likelihood of a poissonian pdf:

$$L(\mathbf{g},|,\mathbf{f}) = \prod_m \frac{1}{g_m!} \left(\overline{g_m}\right)^{g_m} \times \exp\left(-\overline{g_m}\right) \tag{6}$$

In Eq. (6), $\overline{g_m} = E\{g_m|\mathbf{f}\}$ is the conditional expectation of $g_m$ events collected by line of sight $m$, given $\mathbf{f}$, the emissivity profile. In practice, the ML algorithm is implemented with an iterative formula, derived from a specific version of the Expectation Maximization algorithm, to converge on the final estimate $\mathbf{f}_{ML}^{(k)}$ of $\mathbf{f}$. Using the same notation as in ref. 69, in terms of the sensitivity $s_n = \sum_m H_{mn}$, the following equation holds:

$$\overline{g_m} = H_{mn} f^n \tag{7}$$

Where the matrix $H$, describing the contribution of each pixel to each detector, is obtained as usual from the layout of the bolometer lines of sight and their etendues.

The iterative formula, to solve the inversion problem by updating the 2D profile until a convergence criterion is satisfied, can be written as:

$$f_n^{(k+1)} = \frac{f_n^{(k)}}{s_n} \left(g_m / H_{mj} f^{j(k)}\right) H_n^m \tag{8}$$

The initial guess can be either uniform or a specific distribution. The former solution is more general and computationally faster but the latter approach has demonstrated to reduce drastically the occurrence of artefacts[71].

The ML approach has the competitive advantage of computing also the covariance matrix $\mathbf{K}_{ML}^{(k)}$ and therefore, for each 2D emissivity distribution, an estimate of the uncertainty associated to each pixel can be provided routinely, without making recourse to delicate and time-consuming Monte Carlo techniques. All the details can be found in ref. 69. It should be noted that the capability of the ML to provide confidence intervals in the solutions is due also to the fact that the algorithm does not include any mathematical regularization. The only imposed constraint, common to all the tomographic inversion techniques implemented on JET, consists of smoothing the reconstructed tomograms along the magnetic surfaces. Examples of reconstructions are reported in Fig. 2. It should be also emphasised that, with adequate specific refinements, the ML algorithm can be modified to take into account the most important sources of errors on the measurements, such as outliers, faulty or missing detectors and uncertainties in the estimates of the magnetic topology[72,73].

Notwithstanding its competitive advantages, the ML tomography shares with the other inversion methods an important drawback; its computational times are not compatible with real-time applications. Completing the calculations for a single time slice typically requires several seconds, a time span far too long for the task of predicting the occurrence of disruptions. Acceleration techniques are under investigation but the deployment of the ML tomography in feedback remains far in the future[74]. The ML

tomography algorithms are therefore utilised in the present work only as a benchmark.

For real-time deployment, therefore, a simplified algorithm has been developed. The main idea consists of trading spatial resolution for speed. To this end, the lines of sights of the two bolometric cameras have been divided in three macro-views each (see Supplementary Fig. 3). Their intersection results in 8 macro-pixels, which cover the entire plasma cross section and have enough spatial resolution to allow identifying all the major radiation anomalies detected on JET. To obtain the local emissivity one has to invert the following systems of equations:

$$P_{H_1} = R_{Div}F_{Div,H_1} + R_{LFB}F_{LFB,H_1}$$

$$P_{H_2} = R_{HFL}F_{HFL,H_2} + R_{Core}F_{Core,H_2} + R_{LFR}F_{LFR,H_2}$$

$$P_{H_3} = R_{HFT}F_{HFT,H_3} + R_{Top}F_{Top,H_3} + R_{LFT}F_{LFT,H_3}$$

$$P_{V_1} = R_{LFB}F_{LFB,V_1} + R_{LFR}F_{LFR,V_1} + R_{LFT}F_{LFT,V_1}$$

$$P_{V_2} = R_{Div}F_{Div,V_2} + R_{Core}F_{Core,V_2} + R_{Top}F_{Top,V_2}$$

$$P_{V_3} = R_{HFT}F_{HFT,V_3} + R_{HFL}F_{HFL,V_3}$$

where the subscripts indicate the macro-pixels depicted in Supplementary Fig. 3 (HFL = high filed low; HFT = high field top; LFR = low field right; LFT = low field top; LFB = low field bottom; Cor = Core; Top = Top; Div = Divertor). The systems consists of 6 equations with 8 unknowns, which can be complemented with 8 weak constraints, by imposing that the emissivities are non-negative. The inversion of this set of equations has been performed with a non-negative least square minimisation method, as reported in ref. [75]. The inversion software has proved to always converge and the obtained emissivity of the 8 macro-pixels are typically within ±20% of what would be obtained with the more sophisticated ML method. The results are always provided in less than 50 μs. Consequently, the fast tomographic approach just described has the accuracy and spatial resolution to identify all the major radiation anomalies affecting JET plasmas and its computational times are more than compatible with deployment in feedback. The 8 macro-pixels of Supplementary Fig. 3 are combined to obtain the four main regions depicted in Fig. 3.

### The estimate of the macro-pixels' internal energy

To apply the model formulated in Eq. (1), the internal energy of each macro-pixel must be determined with sufficient accuracy. The approach adopted, to obtain the results reported in the present work, is described in this section.

The plasma thermal energy per unit volume can be written as:

$$u = \frac{p_i + p_e}{\gamma - 1} = \frac{n_i T_i + n_e T_e}{\gamma - 1} \tag{9}$$

Where the subscript $e$ indicates the electrons and $i$ the ions. Assuming $T_e = T_i = T$ and deuterium fully ionized plasmas ($n_e = n_i = n$ e $\gamma = 5/3$), one obtains:

$$u = 3nT \tag{10}$$

The energy inside a certain volume $V$ can then be calculated as:

$$E_V = \int_V 3nTdV \tag{11}$$

To determine the plasma energy in a given region, one can therefore derive the magnetic topology from an equilibrium code and the temperature and density profiles from the Thomson scattering. Under the usual assumption that the magnetic surfaces are isobars, it is possible to calculate the temperature, density and pressure fields as shown in the example reported in Supplementary Fig. 4. From these maps, the energy in the regions of interest can be approximated with discrete integrals of the form:

$$E_k = \int_{V_k} 3nTdV = 6\pi \sum \sum 2n_{e,i,j}T_{e,i,j}R_{i,j}dRdZ \tag{12}$$

Where $n_{e,i,j}$ and $T_{e,i,j}$ are the density and temperature of the pixel, whose barycentre has the coordinates ($i, j$). $dR$ e $dZ$ are the dimensions along $R$ and $Z$ and $R_{i,j}$ is the major radius of the corresponding pixel barycentre.

Unfortunately on JET there is no equilibrium real-time reconstruction code routinely available. A fall-back solution consists of fitting the HRTS profiles in the region from $R = 3$ to $R = 3.9$ m with a second-order polynomial. For new discharges, the energy in each macro-pixel is then determined by averaging the HRTS profiles in its volume. Supplementary Table 2 reports an estimate of the errors in the macro-pixels' energy committed by adopting this approach. It is worth pointing out that this choice keeps the number of diagnostics required the entire methodology to a minimum since the HRTS is also the system used for the detection of hollowness and edge cooling.

### The adaptive estimates of the probability and time to anomaly

In the present work, the probabilities of the various anomalies are obtained with basically the same mathematical procedure. The amplitude difference of the relevant signals between disruptive and safe examples are fitted with a sigmoid function, to obtain an output between 0 and 1. The fits are then updated with adaptive procedures, implemented with specific neural networks as described in the following. The delicate aspect is the determination of the thresholds, above which alarms are to be raised.

In the case of the locked mode, the approach of ref. [20] has been followed. The critical value of the dimensionless Locked Mode signal (the locked mode amplitude divided by the plasma current $LM_N$) is linked to the internal inductance ($l_i$) by the equation:

$$LM_{N,threshold} = al_i^b \tag{13}$$

The sigmoid function to determine the probability of the plasma disrupting is:

$$p_{LM} = e^{d_{1LM}(LM_N - LM_{N,threshold}) + d_{2LM}} / (1 + e^{d_{1LM}(LM_N - LM_{N,threshold}) + d_{2LM}}) \tag{14}$$

At the beginning of a new tokamak experimental campaign, no data is available and the therefore the algorithm starts with the values reported in ref. [20]. Equation (14) is then updated with the incoming data with a dedicated neural network, using the cross entropy as loss function.

In the case of the temperature hollowness, the following treatment is implemented. The probability of the temperature profile becoming hollow is determined by the sigmoid:

$$p_{ETH} = e^{d_{1ETH}(ETH - ETH_{threshold}) + d_{2ETH}} / (1 + e^{d_{1ETH}(ETH - ETH_{threshold}) + d_{2ETH}}) \tag{15}$$

Since the indicator is a form factor (and not an absolute quantity), the numerical values of the ETH threshold and the other parameters in (15) are derived directly from ref. [20]. With the evolution of the campaign, the sigmoid function is then updated with a dedicated neural network, using the cross entropy as loss function.

As already mentioned, the edge cooling (EC) is detected using the cumulative distribution function (CDF) of the normalised electron temperature profile and by calculating a "edge radius" $\rho_{98}$ as the radius containing the 98% of the CDF. Again, by using a sigmoid function, the probability to have an EC is computed:

$$p_{EC} = e^{d_{1EC}(EC-EC_{threshold})+d_{2EC}} / (1 + e^{d_{1EC}(EC-EC_{threshold})+d_{2EC}}) \qquad (16)$$

The first values of the coefficients $d_1$ and $d_2$ have been derived from the reference paper[22]. The sigmoid function is then updated along the campaign with a dedicated neural network, using the cross entropy as loss function.

In the case of radiation anomalies, a typical classification algorithm would not perform well because any automatic adaptive training set would have a couple of weaknesses. First it would be highly unbalanced, since for most time slices the plasma radiation is not anomalous. Secondly the safe set would unavoidably contain very misleading examples of anomalous patterns classified as safe (due for example to interventions of the control system). An effective way to counteract these problems consists of calculating the probability density function (pdf) of the various cooling factors for both safe and disruptive discharges. So, for each region, the pdfs of "safe" and "(potentially) anomalous" $\Lambda$ are derived. At higher values of $\Lambda$ the two pdfs diverge, because the one of the anomalous patterns becomes larger at larger $\Lambda$. An example for the core radiation is shown in the top plot of Supplementary Fig. 5. Consequently the cumulative density functions (CDF), integrals of the pdfs, tend to separate significantly as shown in the bottom plot of Supplementary Fig. 5. The true positive percentage (or sensitivity) trend is calculated as one minus the anomalous CDF, while the false alarm percentage (or 1 – specificity) is one minus the safe CDF. The results of the present work have been obtained with the following rules for determining the $\Lambda$ thresholds:

1. Search for the threshold that ensure false alarms minor than 0.05% and a true positive larger than false alarms.
2. If no thresholds satisfy the previous rule, the false alarm limit is increased to 0.1%.
3. If no thresholds satisfy the previous rules, the false alarm limit is increased to 1%.

Being based on the CDFs, these threshold are more robust against misclassifications in the training set. Once identified the thresholds, again the probability of a local radiation anomaly is calculated with a sigmoid function:

$$p_\Lambda = e^{d_{1\Lambda}(\Lambda-\Lambda_{threshold})+d_{2\Lambda}} / \left(1 + e^{d_{1\Lambda}(\Lambda-\Lambda_{threshold})+d_{2\Lambda}}\right) \qquad (17)$$

Note that a $\Lambda_{threshold}$ is found for each macro region and an alarm is raised if one of the $p_\Lambda$ exceeds 0.5.

The just described classifiers are able to detect anomalies and label them with reference to the macro-pixels illustrated in Fig. 3. This is of course very valuable information. However, in order to optimise the remedial strategy, the control system would need also an estimate of the time available to undertake a certain action. To this end, three more predictors have been developed to estimate the time to mode locking, hollowness, and edge cooling ($t_{r,LM}$, $t_{r,ETH}$ and $t_{r,EC}$). This aspect is addressed as a supervised regression problem. For each event to forecast, a predictor in the form of a feed-forward neural network is used. The architecture implemented consists of three hidden layers with 20, 10 and 5 neurons respectively. The inputs are: the magnetic toroidal field ($B_t$), the plasma current ($I_p$), the dimensionless Locked Mode (LMN), the internal inductance ($l_i$), the electron temperature and density in the core, middle and edge regions, the electron temperature profile anomaly indicators (ETH and EC), the $\Lambda_i$ in the four regions (Core, HF, LF and Div) and the $\Lambda_{heating}$. The outputs, i.e. the times to the

anomaly, are predicted in the range from 1 ms to 2 s (2 s is considered the time horizon for reliably predicting the earliest events). The loss function is the mean square error of the differences between the predictions and the actual occurring times of the anomalies, weighting more small times to compensate the fact that at small times there are less data (for example, from 1 ms to 10 ms there are 10 points per pulse, while from 100 ms to 1 s there are 900 points). The implemented loss function can therefore be written as:

$$loss_{W,MSE,j} = \frac{\sum W_i \left(\log_{10} t_{r,j,i} - \log_{10} y_{r,j,i}\right)^2}{\sum W_i} \qquad (18)$$

Where:

$$W_i = \frac{1}{y_{r,j,i}} \qquad (19)$$

While $t_{r,j,i}$ and $y_{r,j,i}$ are the predicted and the target time to anomaly $j$ of the $i$-th observation in region $r$. An example for the prediction quality that can be achieved with this approach is shown in Supplementary Fig. 6.

## JET tools for radiation control and examples proving the added value of the proposed strategy

On JET with the metallic wall, the influx of high Z impurities and the consequent radiation emission have proved to be fundamental aspects influencing plasma stability and performances. The most harmful impurities are typically W sputtered by the divertor, Ni and copper, whose source are the neutral beam duct scrapers, and Ni coming from the ICRH antennas[27]. To reduce the influxes of these metal atoms, it is necessary to carefully control the plasma edge parameters, by fine tuning the density, the temperature and the ELM behaviour. The main actuators available on JET for this purpose are gas injection, the pellets of frozen hydrogen isotopes and vertical kicks (rapid movements of the plasma vertical position obtained by changing the radial magnetic field)[27]. Unfortunately, vertical kicks are not compatible with high current because of the risk of vertical displacement events. In its turn, the level of gas injection required to control the ELMs is typically so high as to compromise good confinement. The best strategy devised is therefore a suitable combination of gas fuelling and pacing pellets[23]. However, the fine tuning of these actuator has proved quite delicate, particularly because it is not straightforward to transfer the recipes optimised for deuterium to the other isotopic composition (full tritium and DT). Increasing the plasma mass indeed tends to results in slower ELMs dynamics with longer ELM-free periods, which can result in stronger penetration of the impurities. This problem is compounded by the different power deposition of the T beams, more peaked toward the edge, and the different behaviour of the tritium gas injection modules[23]. However, the indicators, tools and techniques developed in the present work are equally effective with all isotopic compositions. This fact proves their generality and is quite encouraging in view of their application to the next generation of devices.

For mitigation purposes, the main signal used by JET control system is the amplitude of the locked mode. As already discussed, to trigger a control plasma termination, excessive radiation is typically the earliest precursors. The radiation emission is therefore monitored by JET both real-time control systems: the Real Time Central Controller (RTCC) and Plasma Event TRiggering for Alarm (PETRA)[27]. RTCC relies on two main indicators related to the radiation emission: the radiation fraction (the total emission measured by the bolometers divided by the input power) and various peaking factors (the ratio of bolometric central and peripheral chords)[23]. In PETRA a convolutional neural network, trained with JET traditional tomographic inversion method, is

deployed to monitor the radiation in two macro-regions in the centre and the edge[23]. All these systems allowed carrying out the DTE2 campaign quite effectively but provided performances, which are clearly insufficient in the perspective of ITER and DEMO, as documents by the statistics reported in Table 1.

All these JET metrics are clearly outperformed by the indicators developed in the present work. On the basis of the improved indicators and predictions JET control system would be provided with much better information. Supplementary Table 3 reports a comparison of the alarms detected by the indicators developed in the present work and the actions taken from the JET system. A set of 286 discharges has been analysed, in which the actions undertaken by RTCC and PETRA are clear and well-documented. Inspection of the top parts of the table reveals that, in JET safe discharges, when JET control system does not intervene also the proposed tools do not launch an alarm. However, the developed predictors would intervene in the disruptive discharges not mitigated by JET, showing the capability not to miss practically any disruption. In case of successful actions undertaken by RTCC and PETRA, the warning times provided by the indicators developed in the present work typically are much longer (see Supplementary Table 4). They would have therefore allowed implementing avoidance strategies in a good percentage of cases and better prevention and mitigation actions almost always.

The feedback schemes following the logic in the present work have not been implemented systematically on JET yet but there is ample evidence that they could be effective and results in significant improvements of JET performances in terms of disruption handling. Some representative case are reported in the following.

One example of recovered radiation anomaly in the core is shown in Supplementary Fig. 7. A minor core radiation/hollowness anomaly is observed at $t \sim 9.5$ s, which recovers probably due to high input power that avoids a critical electron temperature profile, giving the plasma enough energy to expel the impurities. A second most intense core radiation is observed from $t = 10.8$ s ($\Lambda_{core} > 1$). This excessive emission leads to the fast cooling of the plasma and the hollowing of the electron temperature profile. However, thanks to the increase of the gas rate a $t = 11$ s, the plasma transits from H to L mode, ensuring a larger impurity transport from the core to the edge. This, together with enough input power to reheat the core, allows recovering a centre peaked temperature profile, avoiding the triggering of macroscopic MHD instabilities. The probabilistic maps of the reporting the level of hollowness and edge cooling vs the λ indicators show that the cause of the temperature anomaly is excessive radiation in the core (see Supplementary Fig. 8).

An interesting example of anomaly in the edge radiation is reported in Supplementary Fig. 9. In this case, JET control system does not undertake any remedial action. The developed indicators detect a MARFE at $t = 15.973$ s. An edge cooling alarm is triggered at $t = 16.083$ s and a locked mode is detected at $t = 16.187$ s. The MARFE is clearly seen also in the frames of the visible cameras, as shown in Supplementary Fig. 10. This sequence of events suggests that with adequate actuators sufficient margins would have been available to avoid this disruption. Moreover, it is not an uncommon occurrence on JET with the metallic wall that MARFEs last for much longer periods than the case of discharge 95993.

## Data availability
Data sets generated during the current study are available from the corresponding author on request.

## Code availability
Codes used during the current study are available from the corresponding author on request.

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

## Acknowledgements

This work has been carried out within the framework of the EUROfusion Consortium, funded by the European Union via the Euratom Research and Training Programme (Grant Agreement No. 101052200—EUROfusion). Views and opinions expressed are however those of the author(s) only and do not necessarily reflect those of the European Union or the European Commission. Neither the European Union nor the European Commission can be held responsible for them. The authors appreciate the work of C. Challis, D. Frigione, L. Garzotti, J. Hobirk, A. Kappatou, E. Lerche, F. Rimini, D. Van Ester in organising and conducting the DT campaign.

## Author contributions

Andrea Murari conceptualised and supervised the entire work and wrote the paper. Riccardo Rossi performed the analysis and wrote the codes. Teddy Craciunescu supervised the analyses related to the tomographic reconstruction. Jesus Vega supervised the machine learning algorithms. Michela Gelfusa supervised the entire project.

## Competing interests

The authors declare no competing interests.

## Additional information

## JET Contributors

J. Mailloux[6], N. Abid[6], K. Abraham[6], P. Abreu[7], O. Adabonyan[6], P. Adrich[8], V. Afanasev[9], M. Afzal[6], T. Ahlgren[10], L. Aho-Mantila[11], N. Aiba[12], M. Airila[11], M. Akhtar[6], R. Albanese[13], M. Alderson-Martin[6], D. Alegre[5], S. Aleiferis[14], A. Aleksa[6], A. G. Alekseev[15], E. Alessi[16], P. Aleynikov[17], J. Algualcil[18], M. Ali[6], M. Allinson[6], B. Alper[6], E. Alves[7], G. Ambrosino[13], R. Ambrosino[13], V. Amosov[19], E. Andersson Sundén[20], P. Andrew[17], B. M. Angelini[21], C. Angioni[22], I. Antoniou[6], L. C. Appel[6], C. Appelbee[6], S. Aria[6], M. Ariola[13], G. Artaserse[21], W. Arter[6], V. Artigues[22], N. Asakura[12], A. Ash[6], N. Ashikawa[23], V. Aslanyan[24], M. Astrain[25], O. Asztalos[26], D. Auld[6], F. Auriemma[1], Y. Austin[6], L. Avotina[27], E. Aymerich[28], A. Baciero[5], F. Bairaktaris[29], J. Balbin[30], L. Balbinot[1], I. Balboa[6], M. Balden[22], C. Balshaw[6], N. Balshaw[6], V. K. Bandaru[22], J. Banks[6], Yu. F. Baranov[6], C. Barcellona[31], A. Barnard[6], M. Barnard[6], R. Barnsley[17], A. Barth[6], M. Baruzzo[21], S. Barwell[6], M. Bassan[17], A. Batista[7], P. Batistoni[21], L. Baumane[27], B. Bauvir[17], L. Baylor[32], P. S. Beaumont[6], D. Beckett[6], A. Begolli[6], M. Beidler[32], N. Bekris[33,34], M. Beldishevski[6], E. Belli[35], F. Belli[21], É. Belonohy[6], M. Ben Yaala[36], J. Benayas[6], J. Bentley[6], H. Bergsåker[37], J. Bernardo[7], M. Bernert[22], M. Berry[6], L. Bertalot[17], H. Betar[38], M. Beurskens[39], S. Bickerton[6], B. Bieg[40], J. Bielecki[41], A. Bierwage[12], T. Biewer[32], R. Bilato[22], P. Bílková[42], G. Birkenmeier[22], H. Bishop[6], J. P. S. Bizarro[7], J. Blackburn[6], P. Blanchard[43], P. Blatchford[6], V. Bobkov[22], A. Boboc[6], P. Bohm[42], T. Bohm[44], I. Bolshakova[45], T. Bolzonella[1], N. Bonanomi[22], D. Bonfiglio[1], X. Bonnin[17], P. Bonofiglo[46], S. Boocock[6], A. Booth[6], J. Booth[6], D. Borba[7,33], D. Borodin[47], I. Borodkina[42,47], C. Boulbe[48], C. Bourdelle[30], M. Bowden[6], K. Boyd[6], I. Božičević Mihalić[49], S. C. Bradnam[6], V. Braic[50], L. Brandt[51], R. Bravanec[52], B. Breizman[53], A. Brett[6], S. Brezinsek[47], M. Brix[6], K. Bromley[6], B. Brown[6], D. Brunetti[6,16], R. Buckingham[6], M. Buckley[6], R. Budny[6], J. Buermans[54], H. Bufferand[30], P. Buratti[21], A. Burgess[6], A. Buscarino[31], A. Busse[6], D. Butcher[6], E. de la Cal[5], G. Calabrò[55], L. Calacci[3], R. Calado[7], Y. Camenen[56], G. Canal[57], B. Cannas[28], M. Cappelli[21], S. Carcangiu[28], P. Card[6], A. Cardinali[21], P. Carman[6], D. Carnevale[3], M. Carr[6], D. Carralero[5], L. Carraro[1], I. S. Carvalho[7], P. Carvalho[7], I. Casiraghi[58], F. J. Casson[6], C. Castaldo[21], J. P. Catalan[18], N. Catarino[7], F. Causa[16], M. Cavedon[22], M. Cecconello[20], C. D. Challis[6], B. Chamberlain[6], C. S. Chang[46], A. Chankin[22], B. Chapman[6,59], M. Chernyshova[60],

A. Chiariello[13], P. Chmielewski[60], A. Chomiczewska[60], L. Chone[61], G. Ciraolo[30], D. Ciric[6], J. Citrin[62], Ł. Ciupinski[63], M. Clark[6], R. Clarkson[6], C. Clements[6], M. Cleverly[6], J. P. Coad[6], P. Coates[6], A. Cobalt[6], V. Coccorese[13], R. Coelho[7], J. W. Coenen[47], I. H. Coffey[64], A. Colangeli[21], L. Colas[30], C. Collins[32], J. Collins[6], S. Collins[6], D. Conka[27], S. Conroy[20], B. Conway[6], N. J. Conway[6], D. Coombs[6], P. Cooper[6], S. Cooper[6], C. Corradino[31], G. Corrigan[6], D. Coster[22], P. Cox[6], T. Craciunescu[4], S. Cramp[6], C. Crapper[6], D. Craven[6], R. Craven[6], M. Crialesi Esposito[51], G. Croci[58], D. Croft[6], A. Croitoru[4], K. Crombé[54,65], T. Cronin[6], N. Cruz[7], C. Crystal[35], G. Cseh[26], A. Cufar[66], A. Cullen[6], M. Curuia[67], T. Czarski[60], H. Dabirikhah[6], A. Dal Molin[58], E. Dale[6], P. Dalgliesh[6], S. Dalley[6], J. Dankowski[41], P. David[22], A. Davies[6], S. Davies[6], G. Davis[6], K. Dawson[6], S. Dawson[6], I. E. Day[6], M. De Bock[17], G. De Temmerman[17], G. De Tommasi[13], K. Deakin[6], J. Deane[6], R. Dejarnac[42], D. Del Sarto[38], E. Delabie[32], D. Del-Castillo-Negrete[32], A. Dempsey[68], R. O. Dendy[6,59], P. Devynck[30], A. Di Siena[22], C. Di Troia[21], T. Dickson[6], P. Dinca[4], T. Dittmar[47], J. Dobrashian[6], R. P. Doerner[69], A. J. H. Donné[70], S. Dorling[6], S. Dormido-Canto[71], D. Douai[30], S. Dowson[6], R. Doyle[68], M. Dreval[72], P. Drewelow[39], P. Drews[47], G. Drummond[6], Ph. Duckworth[17], H. Dudding[6,73], R. Dumont[30], P. Dumortier[54], D. Dunai[26], T. Dunatov[49], M. Dunne[22], I. Ďuran[42], F. Durodié[54], R. Dux[22], A. Dvornova[30], R. Eastham[6], J. Edwards[6], Th. Eich[22], A. Eichorn[6], N. Eidietis[35], A. Eksaeva[47], H. El Haroun[6], G. Ellwood[17], C. Elsmore[6], O. Embreus[74], S. Emery[6], G. Ericsson[20], B. Eriksson[20], F. Eriksson[75], J. Eriksson[20], L. G. Eriksson[76], S. Ertmer[47], S. Esquembri[25], A. L. Esquisabel[77], T. Estrada[5], G. Evans[6], S. Evans[6], E. Fable[22], D. Fagan[6], M. Faitsch[22], M. Falessi[21], A. Fanni[28], A. Farahani[6], I. Farquhar[6], A. Fasoli[43], B. Faugeras[48], S. Fazinić[49], F. Felici[43], R. Felton[6], A. Fernandes[7], H. Fernandes[7], J. Ferrand[6], D. R. Ferreira[7], J. Ferreira[7], G. Ferrò[3], J. Fessey[6], O. Ficker[42], A. R. Field[6], A. Figueiredo[7], J. Figueiredo[7,33], A. Fil[6], N. Fil[6,24], P. Finburg[6], D. Fiorucci[1], U. Fischer[34], G. Fishpool[6], L. Fittill[6], M. Fitzgerald[6], D. Flammini[21], J. Flanagan[6], K. Flinders[6], S. Foley[6], N. Fonnesu[21], M. Fontana[43], J. M. Fontdecaba[5], S. Forbes[6], A. Formisano[13], T. Fornal[60], L. Fortuna[31], E. Fortuna-Zalesna[63], M. Fortune[6], C. Fowler[6], E. Fransson[75], L. Frassinetti[37], M. Freisinger[47], R. Fresa[13], R. Fridström[37], D. Frigione[3], T. Fülöp[74], M. Furseman[6], V. Fusco[27], S. Futatani[21], D. Gadariya[78], K. Gál[70], D. Galassi[43], K. Gałązka[60], S. Galeani[3], D. Gallart[79], R. Galvão[57], Y. Gao[47], J. Garcia[30], M. García-Muñoz[80], M. Gardener[6], L. Garzotti[6], J. Gaspar[81], R. Gatto[82], P. Gaudio[3], D. Gear[6], T. Gebhart[32], S. Gee[6], M. Gelfusa[3], R. George[6], S. N. Gerasimov[6], G. Gervasini[16], M. Gethins[6], Z. Ghani[6], M. Gherendi[4], F. Ghezzi[16], J. C. Giacalone[30], L. Giacomelli[16], G. Giacometti[56], C. Gibson[6], K. J. Gibson[73], L. Gil[7], A. Gillgren[75], D. Gin[9], E. Giovannozzi[21], C. Giroud[6], R. Glen[6], S. Glöggler[22], J. Goff[6], P. Gohil[35], V. Goloborodko[83], R. Gomes[7], B. Gonçalves[7], M. Goniche[30], A. Goodyear[6], S. Gore[6], G. Gorini[58], T. Görler[22], N. Gotts[6], R. Goulding[46], E. Gow[6], B. Graham[6], J. P. Graves[43], H. Greuner[22], B. Grierson[46], J. Griffiths[6], S. Griph[6], D. Grist[6], W. Gromelski[60], M. Groth[61], R. Grove[32], M. Gruca[60], D. Guard[6], N. Gupta[6], C. Gurl[6], A. Gusarov[84], L. Hackett[6], S. Hacquin[30,33], R. Hager[46], L. Hägg[20], A. Hakola[11], M. Halitovs[27], S. Hall[6], S. A. Hall[6], S. Hallworth-Cook[6], C. J. Ham[6], D. Hamaguchi[12], M. Hamed[30], C. Hamlyn-Harris[6], K. Hammond[6], E. Harford[6], J. R. Harrison[6], D. Harting[6], Y. Hatano[85], D. R. Hatch[53], T. Haupt[6], J. Hawes[6], N. C. Hawkes[6], J. Hawkins[6], T. Hayashi[12], S. Hazael[6], S. Hazel[6], P. Heesterman[6], B. Heidbrink[86], W. Helou[17], O. Hemming[6], S. S. Henderson[6], R. B. Henriques[7], D. Hepple[6], J. Herfindal[32], G. Hermon[6], J. Hill[6], J. C. Hillesheim[6], K. Hizanidis[29], A. Hjalmarsson[20], A. Ho[62], J. Hobirk[22], O. Hoenen[17], C. Hogben[6], A. Hollingsworth[6], S. Hollis[6], E. Hollmann[69], M. Hölzl[22], B. Homan[48], M. Hook[6], D. Hopley[6], J. Horáček[42], D. Horsley[6], N. Horsten[61], A. Horton[6], L. D. Horton[33,43], L. Horvath[6,73], S. Hotchin[6], R. Howell[6], Z. Hu[58], A. Huber[47], V. Huber[47], T. Huddleston[6], G. T. A. Huijsmans[17], P. Huynh[30], A. Hynes[6], M. Iliasova[9], D. Imrie[6], M. Imríšek[42], J. Ingleby[6], P. Innocente[1], K. Insulander Björk[74], N. Isernia[13], I. Ivanova-Stanik[60], E. Ivings[6], S. Jablonski[60], S. Jachmich[17,33,54], T. Jackson[6], P. Jacquet[6], H. Järleblad[87], F. Jaulmes[42], J. Jenaro Rodriguez[6], I. Jepu[4], E. Joffrin[30], R. Johnson[6], T. Johnson[37], J. Johnston[6], C. Jones[6], G. Jones[6], L. Jones[6], N. Jones[6], T. Jones[6], A. Joyce[6], R. Juarez[18], M. Juvonen[6], P. Kalniņa[27], T. Kaltiaisenaho[11], J. Kaniewski[6], A. Kantor[6], A. Kappatou[22], J. Karhunen[10], D. Karkinsky[6], Yu Kashchuk[88], M. Kaufman[32], G. Kaveney[6], Ye. O. Kazakov[54], V. Kazantzidis[29], D. L. Keeling[6], R. Kelly[6], M. Kempenaars[17], C. Kennedy[6], D. Kennedy[6], J. Kent[6], K. Khan[6], E. Khilkevich[9], C. Kiefer[22], J. Kilpeläinen[61], C. Kim[35], Hyun-Tae Kim[6,33], S. H. Kim[17], D. B. King[6], R. King[6], D. Kinna[6], V. G. Kiptily[6], A. Kirjasuo[11], K. K. Kirov[6], A. Kirschner[47], T. kiviniemi[61], G. Kizane[27], M. Klas[89], C. Klepper[32], A. Klix[34], G. Kneale[6], M. Knight[6], P. Knight[6], R. Knights[6], S. Knipe[6], M. Knolker[35], S. Knott[90], M. Kocan[17], F. Köchl[6], I. Kodeli[66], Y. Kolesnichenko[83], Y. Kominis[29], M. Kong[6], V. Korovin[72], B. Kos[66], D. Kos[6], H. R. Koslowski[47], M. Kotschenreuther[53], M. Koubiti[56], E. Kowalska-Strzęciwilk[60], K. Koziol[8], A. Krasilnikov[88], V. Krasilnikov[17,19], M. Kresina[6,30], K. Krieger[22], N. Krishnan[6], A. Krivska[54], U. Kruezi[17], I. Książek[91], A. B. Kukushkin[15], H. Kumpulainen[61], T. Kurki-Suonio[61], H. Kurotaki[12], S. Kwak[39], O. J. Kwon[92], L. Laguardia[16], E. Lagzdina[27], A. Lahtinen[10], A. Laing[6], N. Lam[6], H. T. Lambertz[47], B. Lane[6], C. Lane[6], E. Lascas Neto[43], E. Łaszyńska[60], K. D. Lawson[6], A. Lazaros[29], E. Lazzaro[16], G. Learoyd[6], Chanyoung Lee[93], S. E. Lee[85], S. Leerink[61], T. Leeson[6], X. Lefebvre[6], H. J. Leggate[68], J. Lehmann[6], M. Lehnen[17], D. Leichtle[34,94], F. Leipold[17], I. Lengar[66], M. Lennholm[6,76], E. Leon Gutierrez[5], B. Lepiavko[83], J. Leppänen[11], E. Lerche[54], A. Lescinskis[27], J. Lewis[6], W. Leysen[84], L. Li[47], Y. Li[47], J. Likonen[11], Ch. Linsmeier[47], B. Lipschultz[73], X. Litaudon[30,33], E. Litherland-Smith[6], F. Liu[30,33], T. Loarer[30], A. Loarte[17], R. Lobel[6], B. Lomanowski[32], P. J. Lomas[6], J. M. López[25], R. Lorenzini[1], S. Loreti[21], U. Losada[5], V. P. Loschiavo[13], M. Loughlin[17], Z. Louka[6], J. Lovell[32], T. Lowe[6], C. Lowry[6,76], S. Lubbad[6], T. Luce[17], R. Lucock[6], A. Lukin[95], C. Luna[96], E. de la Luna[5],

M. Lungaroni[3], C. P. Lungu[4], T. Lunt[22], V. Lutsenko[83], B. Lyons[35], A. Lyssoivan[54], M. Machielsen[43], E. Macusova[42], R. Mäenpää[61], C. F. Maggi[6], R. Maggiora[97], M. Magness[6], S. Mahesan[6], H. Maier[22], R. Maingi[46], K. Malinowski[60], P. Manas[22,56], P. Mantica[16], M. J. Mantsinen[98], J. Manyer[79], A. Manzanares[99], Ph. Maquet[17], G. Marceca[43], N. Marcenko[88], C. Marchetto[100], O. Marchuk[47], A. Mariani[16], G. Mariano[21], M. Marin[62], M. Marinelli[3], T. Marković[42], D. Marocco[21], L. Marot[36], S. Marsden[6], J. Marsh[6], R. Marshall[6], L. Martellucci[3], A. Martin[6], A. J. Martin[6], R. Martone[13], S. Maruyama[17], M. Maslov[6], S. Masuzaki[23], S. Matejcik[89], M. Mattei[13], G. F. Matthews[6], D. Matveev[47], E. Matveeva[42], A. Mauriya[7], F. Maviglia[13], M. Mayer[22], M.-L. Mayoral[6,70], S. Mazzi[56], C. Mazzotta[21], R. McAdams[6], P. J. McCarthy[90], K. G. McClements[6], J. McClenaghan[35], P. McCullen[6], D. C. McDonald[6], D. McGuckin[6], D. McHugh[6], G. McIntyre[6], R. McKean[6], J. McKehon[6], B. McMillan[59], L. McNamee[6], A. McShee[6], A. Meakins[6], S. Medley[6], C. J. Meekes[62,101], K. Meghani[6], A. G. Meigs[6], G. Meisl[22], S. Meitner[32], S. Menmuir[6], K. Mergia[14], S. Merriman[6], Ph. Mertens[47], S. Meshchaninov[19], A. Messiaen[54], R. Michling[17], P. Middleton[6], D. Middleton-Gear[6], J. Mietelski[41], D. Milanesio[97], E. Milani[3], F. Militello[6], A. Militello Asp[6], J. Milnes[6], A. Milocco[58], G. Miloshevsky[102], C. Minghao[6], S. Minucci[55], I. Miron[4], M. Miyamoto[103], J. Mlynář[42,104], V. Moiseenko[72], P. Monaghan[6], I. Monakhov[6], T. Moody[6], S. Moon[37], R. Mooney[6], S. Moradi[54], J. Morales[30], R. B. Morales[6], S. Mordijck[105], L. Moreira[6], L. Morgan[6], F. Moro[21], J. Morris[6], K.-M. Morrison[6], L. Msero[13,36], D. Moulton[6], T. Mrowetz[6], T. Mundy[6], M. Muraglia[56], A. Murari[1,33], A. Muraro[16], N. Muthusonai[6], B. N'Konga[48], Yong-Su Na[93], F. Nabais[7], M. Naden[6], J. Naish[6], R. Naish[6], F. Napoli[21], E. Nardon[30], V. Naulin[87], M. F. F. Nave[7], I. Nedzelskiy[8], G. Nemtsev[19], V. Nesenevich[9], I. Nestoras[6], R. Neu[22], V. S. Neverov[15], S. Ng[6], M. Nicassio[6], A. H. Nielsen[87], D. Nina[7], D. Nishijima[106], C. Noble[6], C. R. Nobs[6], M. Nocente[58], D. Nodwell[6], K. Nordlund[10], H. Nordman[13], R. Normanton[6], J. M. Noterdaeme[22], S. Nowak[16], E. Nunn[6], H. Nyström[37], M. Oberparleiter[75], B. Obryk[41], J. O'Callaghan[6], T. Odupitan[6], H. J. C. Oliver[6,53], R. Olney[6], M. O'Mullane[107], J. Ongena[54], E. Organ[6], F. Orsitto[13], J. Orszagh[89], T. Osborne[35], R. Otin[6], T. Otsuka[108], A. Owen[6], Y. Oya[109], M. Oyaizu[12], R. Paccagnella[1], N. Pace[6], L. W. Packer[6], S. Paige[6], E. Pajuste[27], D. Palade[4], S. J. P. Pamela[6], N. Panadero[5], E. Panontin[58], A. Papadopoulos[29], G. Papp[22], P. Papp[89], V. V. Parail[6], C. Pardanaud[56], J. Parisi[6,110], F. Parra Diaz[110], A. Parsloe[6], M. Parsons[32], N. Parsons[6], M. Passeri[3], A. Patel[6], A. Pau[43], G. Pautasso[22], R. Pavlichenko[72], A. Pavone[39], E. Pawelec[91], C. Paz Soldan[111], A. Peacock[6,76], M. Pearce[6], E. Peluso[3], C. Penot[17], K. Pepperell[6], R. Pereira[7], T. Pereira[7], E. Perelli Cippo[16], P. Pereslavtsev[108], C. Perez von Thun[60], V. Pericoli[60], D. Perry[6], M. Peterka[42], P. Petersson[37], G. Petravich[26], N. Petrella[6], M. Peyman[6], M. Pillon[21], S. Pinches[17], G. Pintsuk[47], W. Pires de Sá[57], A. Pires dos Reis[57], C. Piron[21], L. Pionr[1,112], A. Pironti[13], R. Pitts[17], K. L. van de Plassche[62], N. Platt[6], V. Plyusnin[7], M. Podesta[46], G. Pokol[26], F. M. Poli[46], O. G. Pompilian[4], S. Popovichev[6], M. Poradziński[60], M. T. Porfiri[21], M. Porkolab[24], C. Porosnicu[4], M. Porton[6], G. Poulipoulis[113], I. Predebon[1], G. Prestopino[3], C. Price[6], D. Price[6], M. Price[6], D. Primetzhofer[20], P. Prior[6], G. Provatas[49], G. Pucella[21], P. Puglia[43], K. Purahoo[6], I. Pusztai[74], O. Putignano[58], T. Pütterich[22], A. Quercia[13], E. Rachlew[74], G. Radulescu[32], V. Radulovic[66], M. Rainford[6], P. Raj[34], G. Ralph[6], G. Ramogida[21], D. Rasmussen[32], J. J. Rasmussen[87], G. Rattá[5], S. Ratynskaia[114], M. Rebai[16], D. Réfy[26], R. Reichle[17], M. Reinke[32], D. Reiser[47], C. Reux[30], S. Reynolds[6], M. L. Richiusa[6], S. Richyal[6], D. Rigamonti[16], F. G. Rimini[6], J. Risner[32], M. Riva[21], J. Rivero-Rodriguez[80], C. M. Roach[6], R. Robins[6], S. Robinson[6], D. Robson[6], R. Rodionov[88], P. Rodrigues[7], M. Rodriguez Ramos[110], P. Rodriguez-Fernandez[8], F. Romanelli[75], M. Romanelli[6], S. Romanelli[6], J. Romazanov[47], R. Rossi[3], S. Rowe[6], D. Rowlands[6,33], M. Rubel[37], G. Rubinacci[13], G. Rubino[55], L. Ruchko[57], M. Ruiz[25], J. Ruiz Ruiz[110], C. Ruset[4], J. Rzadkiewicz[8], S. Saarelma[6], E. Safi[10], A. Sahlberg[20], M. Salewski[87], A. Salmi[11], R. Salmon[6], F. Salzedas[7,115], I. Sanders[6], D. Sandiford[6], B. Santos[7], A. Santucci[21], K. Särkimäki[74], R. Sarwar[6], I. Sarychev[6], O. Sauter[43], P. Sauwan[18], N. Scapin[51], F. Schluck[47], K. Schmid[22], S. Schmuck[16], M. Schneider[17], P. A. Schneider[22], D. Schwörer[68], G. Scott[6], M. Scott[6], D. Scraggs[6], S. Scully[6], M. Segato[6], Jaemin Seo[93], G. Sergienko[47], M. Sertoli[6], S. E. Sharapov[6], A. Shaw[6], H. Sheikh[6], U. Sheikh[43], A. Shepherd[6], A. Shevelev[9], P. Shigin[17], K. Shinohara[116], S. Shiraiwa[46], D. Shiraki[32], M. Short[6], G. Sias[28], S. A. Silburn[6], A. Silva[7], C. Silva[7], J. Silva[6], D. Silvagni[22], D. Simfukwe[6], J. Simpson[6,61], D. Sinclair[6], S. K. Sipilä[61], A. C. C. Sips[76], P. Sirén[10], A. Sirinelli[17], H. Sjöstrand[20], N. Skinner[6], J. Slater[6], N. Smith[6], P. Smith[6], J. Snell[6], G. Snoep[62], L. Snoj[66], P. Snyder[35], S. Soare[4], E. R. Solano[5], V. Solokha[61], A. Somers[68], C. Sommariva[43], K. Soni[36], E. Sorokovoy[72], M. Sos[42], J. Sousa[7], C. Sozzi[16], S. Spagnolo[1], T. Spelzini[6], F. Spineanu[4], D. Spong[32], D. Sprada[6], S. Sridhar[30], C. Srinivasan[6], G. Stables[6], G. Staebler[35], I. Stamatelatos[14], Z. Stancar[66], P. Staniec[6], G. Stankūnas[117], M. Stead[6], E. Stefanikova[37], A. Stephen[6], J. Stephens[6], P. Stevenson[6], M. Stojanov[6], P. Strand[75], H. R. Strauss[118], S. Strikwerda[6], P. Ström[37], C. I. Stuart[6], W. Studholme[6], M. Subramani[6], E. Suchkov[89], S. Sumida[12], H. J. Sun[6], T. E. Susts[27], J. Svensson[39], J. Svoboda[42], R. Sweeney[24], D. Sytnykov[72], T. Szabolics[26], G. Szepesi[6], B. Tabia[6], T. Tadić[49], B. Tál[22], T. Tala[11], A. Tallargio[6], P. Tamain[30], H. Tan[6], K. Tanaka[23], W. Tang[46], M. Tardocchi[16], D. Taylor[6], A. S. Teimane[27], G. Telesca[60], N. Teplova[9], A. Teplukhina[46], D. Terentyev[84], A. Terra[47], D. Terranova[1], N. Terranova[21], D. Testa[43], E. Tholerus[6,37], J. Thomas[6], E. Thoren[114], A. Thorman[6], W. Tierens[22], R. A. Tinguely[24], A. Tipton[6], H. Todd[6], M. Tokitani[23], P. Tolias[114], M. Tomeš[42], A. Tookey[6], Y. Torikai[119], U. von Toussaint[22], P. Tsavalas[14], D. Tskhakaya[42,120], I. Turner[6], M. Turner[6], M. M. Turner[68], M. Turnyanskiy[6,70], G. Tvalashvili[6], S. Tyrrell[6], M. Tyshchenko[83], A. Uccello[16], V. Udintsev[17], G. Urbanczyk[30], A. Vadgama[6], D. Valcarcel[6], M. Valisa[1], P. Vallejos Olivares[37], O. Vallhagen[74], M. Valovič[6], D. Van Eester[54], J. Varje[61], S. Vartanian[30],

T. Vasilopoulou[14], G. Vayakis[17], M. Vecsei[26], J. Vega[5], S. Ventre[13], G. Verdoolaege[65], C. Verona[3], G. Verona Rinati[3], E. Veshchev[17], N. Vianello[1], E. Viezzer[80], L. Vignitchouk[114], R. Vila[5], R. Villari[21], F. Villone[13], P. Vincenzi[1], I. Vinyar[95], B. Viola[21], A. J. Virtanen[61], A. Vitins[27], Z. Vizvary[6], G. Vlad[21], M. Vlad[4], P. Vondráček[42], P. de Vries[17], B. Wakeling[6], N. R. Walkden[6], M. Walker[6], R. Walker[6], M. Walsh[17], E. Wang[47], N. Wang[6], S. Warder[6], R. Warren[6], J. Waterhouse[6], C. Watts[17], T. Wauters[54], A. Weckmann[37], H. Wedderburn Maxwell[6], M. Weiland[22], H. Weisen[43], M. Weiszflog[20], P. Welch[6], N. Wendler[60], A. West[6], M. Wheatley[6], S. Wheeler[6], A. Whitehead[6], D. Whittaker[6], A. Widdowson[6], S. Wiesen[47], J. Wilkinson[6], J. C. Williams[6], D. Willoughby[6], I. Wilson[6], J. Wilson[6], T. Wilson[6], M. Wischmeier[22], P. Wise[6], G. Withenshaw[6], A. Withycombe[6], D. Witts[6], A. Wojcik-Gargula[41], E. Wolfrum[22], R. Wood[6], C. Woodley[6], R. Woodley[6], B. Woods[6], J. Wright[6], J. C. Wright[24], T. Xu[6], D. Yadikin[75], M. Yajima[23], Y. Yakovenko[83], Y. Yang[17], W. Yanling[47], V. Yanovskiy[42], I. Young[6], R. Young[6], R. J. Zabolockis[27], J. Zacks[6], R. Zagorski[8], F. S. Zaitsev[89], L. Zakharov[10], A. Zarins[27], D. Zarzoso Fernandez[56], K. -D. Zastrow[6], Y. Zayachuk[6], M. Zerbini[21], W. Zhang[22], Y. Zhou[37], M. Zlobinski[47], A. Zocco[39], A. Zohar[66], V. Zoita[4], S. Zoletnik[26], V. K. Zotta[82], I. Zoulias[6], W. Zwingmann[2] & I. Zychor[3]

[6]United Kingdom Atomic Energy Authority, Culham Science Centre, Abingdon, Oxon, UK. [7]Instituto de Plasmas e Fusão Nuclear, Instituto Superior Técnico, Universidade de Lisboa, Lisboa, Portugal. [8]National Centre for Nuclear Research (NCBJ), Otwock-Świerk, Poland. [9]Ioffe Physico-Technical Institute, 26 Politekhnicheskaya, St Petersburg, Russia. [10]University of Helsinki, PO Box 43, Helsinki, Finland. [11]VTT Technical Research Centre of Finland, PO Box 1000, Espoo, Finland. [12]National Institutes for Quantum and Radiological Science and Technology, Naka, Ibaraki, Japan. [13]Consorzio CREATE, Via Claudio 21, Napoli, Italy. [14]NCSR 'Demokritos' 153 10, Agia Paraskevi Attikis, Attica, Greece. [15]NRC Kurchatov Institute, 1 Kurchatov Square, Moscow, Russia. [16]Institute for Plasma Science and Technology, CNR, via R. Cozzi 53, Milano, Italy. [17]ITER Organization, Route de Vinon-sur-Verdon, CS 90 046, 13067 Saint Paul Lez Durance Cedex, Verdon, France. [18]Universidad Nacional de Educacion a Distancia, Dept Ingn Energet, Calle Juan del Rosal 12, Madrid, Spain. [19]Troitsk Insitute of Innovating and Thermonuclear Research (TRINITI), Troitsk, Moscow Region, Russia. [20]Department of Physics and Astronomy, Uppsala University, Uppsala, Sweden. [21]Dip.to Fusione e Tecnologie per la Sicurezza Nucleare, ENEA C. R. Frascati, via E. Fermi 45, Frascati (Roma), Italy. [22]Max-Planck-Institut für Plasmaphysik, Garching, Germany. [23]National Institute for Fusion Science, Oroshi, Toki, Gifu, Japan. [24]MIT Plasma Science and Fusion Center, Cambridge, MA, USA. [25]Universidad Politécnica de Madrid, Grupo I2A2, Madrid, Spain. [26]Centre for Energy Research, POB 49, Budapest, Hungary. [27]University of Latvia, 19 Raina Blvd., Riga, LV, Latvia. [28]Department of Electrical and Electronic Engineering, University of Cagliari, Piazza d'Armi, Cagliari, Italy. [29]National Technical University of Athens, Iroon Politechniou 9, 157 73 Zografou, Athens, Greece. [30]CEA, IRFM, Saint Paul Lez Durance, France. [31]Dipartimento di Ingegneria Elettrica Elettronica e Informatica, Università degli Studi di Catania, Catania, Italy. [32]Oak Ridge National Laboratory, Oak Ridge, TN, USA. [33]EUROfusion Programme Management Unit, Culham Science Centre, Culham, UK. [34]Karlsruhe Institute of Technology, PO Box 3640 Karlsruhe, Germany. [35]General Atomics, PO Box 85608 San Diego, CA, USA. [36]Department of Physics, University of Basel, Basel, Switzerland. [37]Fusion Plasma Physics, EECS, KTH Royal Institute of Technology, Stockholm, Sweden. [38]Institut Jean Lamour, UMR 7198, CNRS-Université de Lorraine, Nancy, France. [39]Max-Planck-Institut für Plasmaphysik, Teilinsitut Greifswald, Greifswald, Germany. [40]Maritime University of Szczecin Facultcodey of Marine Engineering, Waly Chrobrego 1-2, Szczecin, Poland. [41]Institute of Nuclear Physics, Radzikowskiego 152, Kraków, Poland. [42]Institute of Plasma Physics of the CAS, Za Slovankou 1782/3, 182 00 Praha 8, Prague, Czech Republic. [43]Ecole Polytechnique Fédérale de Lausanne (EPFL), Swiss Plasma Center (SPC), Lausanne, Switzerland. [44]University of Wisconsin-Madison, Madison, WI, USA. [45]Magnetic Sensor Laboratory, Lviv Polytechnic National University, Lviv, Ukraine. [46]Princeton Plasma Physics Laboratory, James Forrestal Campus, Princeton, NJ, USA. [47]Forschungszentrum Jülich GmbH, Institut für Energie- und Klimaforschung, Plasmaphysik, Jülich, Germany. [48]Université Cote d'Azur, CNRS, Inria, LJAD, Parc Valrose, 06108 Nice Cedex 02, Valrose, France. [49]Ruđer Bošković Institute, Bijenička 54, Zagreb, Croatia. [50]The National Institute for Optoelectronics, Magurele-Bucharest, Romania. [51]Mechanics, SCI, KTH SE-100 44, Stockholm, Sweden. [52]Fourth State Research, 503 Lockhart Dr, Austin, TX, USA. [53]University of Texas at Austin, Institute for Fusion Studies, Austin, TX, USA. [54]Laboratory for Plasma Physics LPP-ERM/KMS, Brussels, Belgium. [55]University of Tuscia, DEIM, Via del Paradiso 47, Viterbo, Italy. [56]Aix-Marseille University, CNRS, PIIM, UMR 7345, Marseille, France. [57]Instituto de Física, Universidade de São Paulo, Rua do Matão Travessa R Nr.187, CEP 05508-090 Cidade Universitária, São Paulo, Brasil. [58]University of Milano-Bicocca, Piazza della Scienza 3, Milano, Italy. [59]Centre for Fusion, Space and Astrophysics, University of Warwick, Coventry, UK. [60]Institute of Plasma Physics and Laser Microfusion, Hery 23, Warsaw, Poland. [61]Aalto University, PO Box 141 Aalto, Finland. [62]FOM Institute DIFFER, Eindhoven, The Netherlands. [63]Warsaw University of Technology, 02-507, Warsaw, Poland. [64]Astrophysics Research Centre, School of Mathematics and Physics, Queen's University, Belfast, UK. [65]Department of Applied Physics, Ghent University, Ghent, Belgium. [66]Slovenian Fusion Association (SFA), Jozef Stefan Institute, Jamova 39, Ljubljana, Slovenia. [67]The National Institute for Cryogenics and Isotopic Technology, Ramnicu Valcea, Romania. [68]Dublin City University (DCU), Dublin, Ireland. [69]University of California at San Diego, La Jolla, CA, USA. [70]EUROfusion Programme Management Unit, Boltzmannstr. 2, Garching, Germany. [71]UNED, Dpto. Informática y Automática, Madrid, Spain. [72]National Science Center 'Kharkov Institute of Physics and Technology', Akademichna 1, Kharkiv, Ukraine. [73]York Plasma Institute, Department of Physics, University of York, York, UK. [74]Department of Physics, Chalmers University of Technology, Gothenburg, Sweden. [75]Department of Space, Earth and Environment, Chalmers University of Technology, Gothenburg, Sweden. [76]European Commission, Brussels, Belgium. [77]University of Tennessee, Knoxville, TN, USA. [78]Universitat Politècnica de Catalunya, Barcelona, Spain. [79]Barcelona Supercomputing Center, Barcelona, Spain. [80]Universidad de Sevilla, Sevilla, Spain. [81]Aix-Marseille University, CNRS, IUSTI, UMR 7343, Marseille, France. [82]Dipartimento di Ingegneria Astronautica, Elettrica ed Energetica, SAPIENZA Università di Roma, Via Eudossiana 18, Roma, Italy. [83]Institute for Nuclear Research, Prospekt Nauky 47, Kyiv, Ukraine. [84]Studiecentrum voor Kernenergie—Centre d'Etude de l'Energie Nucléaire, Boeretang 200, Mol, Belgium. [85]University of Toyama, Toyama, Japan. [86]University of California, Irvine, California, USA. [87]Department of Physics, Technical University of Denmark, Bldg 309, Kgs Lyngby, Denmark. [88]Institution 'Project Center ITER', Moscow, Russia. [89]Faculty of Mathematics, Department of Experimental Physics, Physics and Informatics Comenius University Mlynska dolina F2, Bratislava, Slovakia. [90]University College Cork (UCC), Cork, Ireland. [91]Institute of Physics, Opole University, Oleska 48, Opole, Poland. [92]Daegu University, Jillyang, Gyeongsan, Gyeongbuk, Republic of Korea. [93]Department of Nuclear Engineering, Seoul National University, Seoul, Republic of Korea. [94]Fusion for Energy Joint Undertaking, Josep Pl. 2, Torres Diagonal Litoral B3, Barcelona, Spain. [95]PELIN LLC, 27a, Gzhatskaya Ulitsa, Saint Petersburg, Russia. [96]Arizona State University, Tempe, AZ, USA. [97]Politecnico di Torino, Corso Duca degli Abruzzi 24, Torino, Italy. [98]ICREA and Barcelona Supercomputing Center, Barcelona, Spain. [99]Universidad Complutense de Madrid, Madrid, Spain. [100]Istituto dei Sistemi Complessi—CNR and Dipartimento di Energia—Politecnico di Torino, C.so Duca degli Abruzzi 24, Torino, Italy. [101]Eindhoven University of Technology, Eindhoven, The Netherlands. [102]Purdue University, 610 Purdue Mall, West Lafayette, IN, USA. [103]Department of Material Science, Shimane University, 1060 Nishikawatsu,

Matsue, Japan. [104]Faculty of Nuclear Sciences and Physical Engineering, Czech Technical University in Prague, Břehová 78/7, 115 19 Praha 1, Prague, Czech Republic. [105]College of William and Mary, Williamsburg, VA, USA. [106]University of California, 1111 Franklin St., Oakland, CA, USA. [107]University of Strathclyde, Glasgow, UK. [108]Kindai University, Higashi-Osaka, Osaka, Japan. [109]Shizuoka University, Shizuoka, Japan. [110]Rudolf Peierls Centre for Theoretical Physics, University of Oxford, Oxford, UK. [111]Columbia University, New York, NY, USA. [112]Dipartimento di Fisica 'G. Galilei', Universita' degli Studi di Padova, Padova, Italy. [113]Space and Plasma Physics, EECS, KTH, Stockholm, Sweden. [114]University of Ioannina, Panepistimioupoli Ioanninon, PO Box 1186, Ioannina, Greece. [115]Universidade do Porto, Faculdade de Engenharia, Porto, Portugal. [116]The University of Tokyo, Kashiwa, Chiba, Japan. [117]Lithuanian Energy Institute, Breslaujos g. 3, Kaunas, Lithuania. [118]HRS Fusion, West Orange, NJ, USA. [119]Ibaraki University Graduate School of Science and Engineering, Mito, Ibaraki, Japan. [120]Technische Universität Wien, Fusion@ÖAW Österreichische Akademie der Wissenschaften (ÖAW), Wien, Austria.

