## [Peer Review File · Nature Communications]

A Control Oriented Strategy of Disruption Prediction to Avoid the Configuration Collapse of Tokamak ReactorsREVIEWER COMMENTS

Reviewer #1 (Remarks to the Author):

This paper talked about the JET way of doing disruption prediction for prevention avoidance and mitigation. The highlight of this paper is its performance and its simplicity. It achieved great performance on multiple campaigns including DT experiments. The method presented is very interesting. It uses multiple physics guided features and an adaptive threshold to rise alarms. Each set of features serves for different purpose. They use LM for mitigation, edge cooling and temperature hollowing for prevention and radiation to internal energy ration for avoidance. The method itself is simple and seen from the results reposted, it's also effective. This is a good reference for developing disruption prediction system.

However, there are a few weak points in this paper I wish the author can address. After that I think this paper is worth publishing in this journal.

1. The features selected in this work is based on many years of operation of JET. Lots of different features have been tried and compared, and those reported in this paper wins. Those features and their performance may be JET specific. Also, the way of extracting these features are JET specific. This significantly limited the presented method to be used in future tokamaks.
2. The STD. of the time of alarm to next instability reported in this paper is very large, indicate this method is not very stable. In fact, a proper intervene experiment may be needed to show that the avoidance and prevention alarm actually legit. Since the STD is very large, it's hard to tell the difference from prevention alarm and avoidance alarm.
3. The method uses only a few feature sets each for a kind of instability causes. It need more evidence to show those would cover the whole plasma operation space. The shots used in the database composed on this study is not very large. I am not sure about the selection logic of this database. Can it cover the whole operational space of JET?
4. In the method section it mentioned a NN regression model to predict the time to LM is build. I did not quite get it. what's the input of this model? Just the features used to produce the alarm? Than why not use those NN to also output the alarm? I believe this would have better performance than just a threshold.
5. The writing of this paper need revising. There are many acronyms that are missing full names. It's very confusing. In that method part, it gives a lot of citations, but also spend lots of text on talking about basic concept. But, actual useful stuff like how those features are extracted are missing especially those used for comparison.

6. The table V, is it a hindsight? Were the alarms raised in different time periods so they are assigned different types? It needs to be more clear and more specific on the logic of how different types of alarms were raised.

Reviewer #2 (Remarks to the Author):

see attached file

Review to the paper
A.Murari, R.Rossi, T.Craciunescu, J.Vega and M.Gelfusa
A Control Oriented Strategy of Disruption Prediction
to Avoid the Configuration Collapse of Tokamak Reactors,

Results of investigation presented in the paper are very important. They related to disruption avoidance and prevention in tokamaks – one of the main barriers on the way to commercial fusion use. A new approach was developed, which allows beforehand determining the time interval remaining before an incoming disruption in real time.

So, publication of the paper is very desirable.

Volume of the paper is high (33 pages), but all chapters are quite appropriate.

As for reviewer, there is necessity of some corrections in the paper. Set of possible corrections is given below (page by page).

Comments and desires

p.1, it is written:

“the nature of the various forms of collapse is investigated”

Comment: yes, for collapses related to Flat Top and Ramp Down disruptions. May be add few words about disruptions at current Ramp Up stage or inform, that there is no such disruptions.

p.2, left

“repulsive coulomb barrier”

necessary

repulsive Coulomb barrier

“the configuration is plagued by macroscopic instabilities”

better destroyed or violated

p.2, right, it is written

“which almost all the internal energy of the plasma is lost to the wall on time scales of milliseconds”.

Word internal here is inaccurate for my mind.

As for reviewer, not internal, but plasma thermal energy for the case of thermal quench (internal energy include thermal energy W_p and energy of plasma current inside plasma volume W_I): plasma thermal energy is $W_p = 3/2 \cdot \int (p_e + p_i) dV$; magnetic energy inside plasma volume is $W_I = L_p \cdot I_p^2 / 2$.

p.3, left

in reactor **grade** devices better: in reactor **scale** devices

It is written:

“The Joint European Tokamak (JET) is the largest tokamak ever operated in the world”

Now it is not so, because of JT-60SA operation...

So, may be better:

The Joint European Tokamak (JET) is **one of** the largest tokamak ever operated in the world

p.4, left

The analysed database consists of **1735** discharges, of which **531** disruptive.

- Another numerals are in Table 1, i.e. (from Table 1: total - **1683** discharges, disruptive – **542**)

It is better use the same number for total and disruptive discharges

p.5

Fig.2

Pulse 94199 - $t = 9.2$
MARFE Radiation

Necessary to add: $t = 9.2$ s

p.8, left, it is written:

“All these emission events are cases of radiation **runaway** instabilities, because the impurities have a radiation function increasing with the inverse of the temperature”

Word “runaway” is often used in application to process of “runaway electrons evolution” during current quench.

So, may be better to write:

All these emission events are cases of radiation **positive feedback** instabilities, because the impurities have a radiation function increasing with the inverse of the temperature

p.8, right, it is written:

“suitable indicator proves to be the ratio of the radiated power divided by the plasma **internal** energy.

This quantity is much more useful than the traditional ratio of radiated power divided by input power

E_p is the plasma internal energy”

Once more: it is better to replace “internal energy” by “plasma thermal energy”

p.10, right

Table III The time intervals between the various anomalies: FT indicates the flat top and RD the ramp down phase of the discharges. Since the time intervals are never negative, large standard deviations are to be considered positive, because they indicate that in many cases the control system would have plenty of time to intervene.

	Mean	Std
$\Delta t_{A,core} - \Delta t_{ETH} (all)$	167	270
$\Delta t_{A,core} - \Delta t_{ETH} (FT)$	139	435

It is necessary to indicate in Table III units of time interval (most likely in **milliseconds**?)

p.13

Table 5, it is written:

Avoided	18.3%	18.0%	0.4%
----------------	-------	-------	------

- All must be equal to sum of FT and RD (18.3 must be equal to 18.0 + 0.4)

The same is for

Prevented	14.9%	3.9%	11.1%
------------------	-------	------	-------

(14.9 must be equal to 3.9 + 11.1)

p.13, left

The quality of such estimates are reported in **figure 6**,

Necessary:

The quality of such estimates are reported in **Figure 6**,

p.14, right, it is written:

“From a methodological perspective, a potentially very significant improvement would consist of state aware tools, particularised for the main phases of the discharge, ramp up, flat top and ramp **of** down of the plasma current”.

May be better without **of**:

“From a methodological perspective, a potentially very significant improvement would consist of state aware tools, particularised for the main phases of the discharge: ramp up, flat top and ramp down of the plasma current”.

p.22, left

Written:

Assuming $T_e = T_i = T$ and deuterium fully ionised plasmas ($n_e = n_i = n$ e $\gamma = 5/3$), one obtains:

Better:

Assuming $T_e = T_i = T$ and deuterium fully ionized plasmas ($n_e = n_i = n$ e $\gamma = 5/3$), one obtains:

instead N_e better n_e

The energy inside a certain volume V can then be calculated as:

$$E_V = \int_V \frac{3}{2} n T dV \quad (\text{M3.3})$$

Is it good?

Recommended version:

Plasma thermal energy inside certain volume V can be calculated as:

$$E_V = \int_V 3nT dV \quad (\text{M3.3})$$

Because

$$E_V = \int_V \frac{3}{2} \cdot (n_e T_e + n_i T_i) dV = \int_V 3nT dV$$

fitting the HRTS profiles in the region from $R = 3$ to $R = 3.9$ with a second order polynomial

Better

fitting the HRTS profiles in the region from $R = 3$ m to $R = 3.9$ m with a second order polynomial

p.23

Table M3.1 Average difference between the estimate of the macro-pixels' energy calculated with equation **M3.3**

May be

Table M3.1 Average difference between the estimate of the macro-pixels' energy calculated with equation **M3.4**

We can see two formulas with the same designation M4.1, i.e.

$$LM_{N,threshold} = a l_i^b \quad \text{M4.1}$$

The sigmoid function to determine the probability of the plasma disrupting is:

$$P_{LM} = \frac{e^{d_{1LM}(LM_N - LM_{N,threshold}) + d_{2LM}}}{1 + e^{d_{1LM}(LM_N - LM_{N,threshold}) + d_{2LM}}} \quad \text{M4.1}$$

Better version for (M4.2):

$$p_{LM} = \frac{x_{LM}}{1 + x_{LM}} \quad (\text{M4.2})$$

where $x_{LM} = e^{d_{1LM} \cdot (LM_N - LM_{N,threshold}) + d_{2LM}}$

(all formulas must be in brackets, i.e. (M4.4), (M4.5), (M4.6), (M4.7))

$$p_{ETH} = \frac{e^{d_{1ETH}(ETH - ETH_{threshold}) + d_{2ETH}}}{1 + e^{d_{1ETH}(ETH - ETH_{threshold}) + d_{2ETH}}} \quad \text{M4.3}$$

May be better

$$p_{ETH} = \frac{x_{ETH}}{1 + x_{ETH}} \quad (\text{M4.3})$$

where $x_{ETH} = e^{d_{1ETH} \cdot (ETH - ETH_{threshold}) + d_{2ETH}}$

The plasma internal energy per unit volume can be written as:

$$u = \frac{p_i + p_e}{\gamma - 1} = \frac{n_i T_i + n_e T_e}{\gamma - 1} \quad (\text{M3.1})$$

It's better to write The **plasma thermal energy** per unit volume...

Because internal energy usually is sum of thermal energy + magnetic energy, connected with plasma current

p.22, Figure M3.1

Let's check data from Fig. M3.1

We have $n_e(0) = 0.42 \cdot 10^{20} \text{ m}^{-3}$, $T_e(0) = 4 \text{ keV}$.

So electron pressure is $p(0) = n_e(0) \cdot T_e(0) = 2.7 \cdot 10^4 \text{ J} \cdot \text{m}^{-3}$. Total pressure assumed as $5.4 \cdot 10^4 \text{ J} \cdot \text{m}^{-3}$

This value of electron pressure is the same as in picture, i.e. $2.7 \cdot 10^4 \text{ J} \cdot \text{m}^{-3}$,
see Fig M3.1 (right, bottom).

We see superscribe is **Energy** [J/m^{-3}]. Here are two mistakes.

Correct version of superscribe is **pressure of electrons** [$\text{J} \cdot \text{m}^{-3}$]

(**pressure**, not **Energy**)

There is no Chapter Summary...

Maybe Summary is the same as Abstract. But Summary usually is more detailed, than
Abstract... with some results of investigation...

p. 30-33, References

Different way of writing names in references, for example:

[15] A. Murari

[16] Murari, A.,

Etc.

It is necessary to use one style...

Reviewer #1 (Remarks to the Author):

This paper talked about the JET way of doing disruption prediction for prevention avoidance and mitigation. The highlight of this paper is its performance and its simplicity. It achieved great performance on multiple campaigns including DT experiments. The method presented is very interesting. It uses multiple physics guided features and an adaptive threshold to rise alarms. Each set of features serves for different purpose. They use LM for mitigation, edge cooling and temperature hollowing for prevention and radiation to internal energy ration for avoidance. The method itself is simple and seen from the results reposted, it's also effective. This is a good reference for developing disruption prediction system.

ANSWER: first, on behalf of my co-authors I would really like to thank the reviewer for the time and efforts spent reviewing our work. We are also very grateful for the positive assessment and the opportunity to review our manuscript. We found all the comments very sensible and appropriate. Consequently we have tried our best to answer all the questions and implement the required modifications. We hope that the new version of the manuscript can now be considered up to the standards of the journal but, in any case, we remain available to implement further modifications if considered indispensable for publication.

However, there are a few weak points in this paper I wish the author can address. After that I think this paper is worth publishing in this journal.

1. The features selected in this work is based on many years of operation of JET. Lots of different features have been tried and compared, and those reported in this paper wins. Those features and their performance may be JET specific. Also, the way of extracting these features are JET specific. This significantly limited the presented method to be used in future tokamaks.

ANSWER: the reviewer is absolutely right that there are strong uncertainties about the diagnostic coverage in future tokamaks. For example, the discussion about the diagnostic suite of DEMO is still going on quite lively. However, we would like to make a few points. First, it is becoming clear that controlling the future tokamak reactor will need good measurements and observers. More importantly for the scope of the present work, we would like to emphasize that we have used only standard diagnostics and simple features exactly to increase the generality of the solutions. The locked mode derived from the magnetics, the temperature profile and bolometry are all diagnostics considered essential also in DEMO. These diagnostics were also all available in JET real time network. The locked mode amplitude is a signal nowadays available in all major tokamaks and should not be an issue in the reactor, which will need a good set of magnetic diagnostics. To interpret the temperature measurements of the Thomson scattering, we implemented only profile indicators, which are immediate to convert from one device to another. The most JET specific indicators are probably those derived from the tomographic inversion of bolometry but we are already implementing the same approach on other European machines without encountering major difficulties. We have also published in the past some preliminary work on transferring these types of indicators and predictors from various devices and we do not see this aspect as an insurmountable difficulty (see for example references 18 and 19 of the manuscript; Murari, A., Rossi, R., Peluso, E., Lungaroni, M., Gaudio, P., Gelfusa, M., ... & Vega, J. (2020). *On the transfer of adaptive predictors between different devices for both mitigation and prevention of disruptions*. Nuclear Fusion, 60(5), 056003.[19] Murari, A., Rossi, R., Lungaroni, M., Baruzzo, M., & Gelfusa, M. (2021). *Stacking of predictors for the automatic classification of disruption types to optimize the*

control logic. Nuclear Fusion, 61(3), 036027). We have summarised all these considerations and more at the beginning of Section M.2, which reads: “*In general, the diagnostics for disruption prediction have to provide more stringent performances than the measurements for the understanding of the physics. Certainly the RAMI (Reliability, Availability, Maintainability, Inspectability) risk control requirements are much more severe, because the needed measurements have to be always available in practically all phases of the experiments [63]. Spatial and time resolutions have also to be carefully considered, since the causes of disruptions are multiple, some quite local and evolving on fast time scales. On top of that, depending on their objectives (mitigation, prevention or avoidance), the various predictors have to rely on different signals. In particular, the measurements for proximity control, for assessing how close to the disruption boundary the plasma state is, have to provide sufficient warning times. For mitigation, the premium of course is on accuracy, since non-mitigated disruptions are particularly bad and probably cannot be tolerated in devices of the size of DEMO [64].*

Whatever the source of information, the adopted metrics to analyse the experimental signals for disruption prediction should present the following desirable properties:

1. Sensitive, in the sense of being capable of detecting the anomalies at an early stage
2. Specific, which means triggering a very small number of false detections
3. Deployable in real-time
4. Easily portable from one tokamak to another

All the diagnostics and derived quantities proposed in the present work satisfy these criteria quite well, as discussed in detail in the following subsections. They are also expected to be available in DEMO and the reactor. The order follows the time proximity to the beginning of the current quench: mitigation, prevention and then avoidance.”

2. The STD. of the time of alarm to next instability reported in this paper is very large, indicate this method is not very stable. In fact, a proper intervene experiment may be needed to show that the avoidance and prevention alarm actually legit. Since the STD is very large, it’s hard to tell the difference from prevention alarm and avoidance alarm.

ANSWER: we fully agree with the reviewer that the STD of the predictions is quite large. However, interpreting correctly this evidence requires a couple of considerations. First we believe that the spread of the predictions is an unavoidable consequence of the nature of the phenomenon and the quality of the measurements available. Indeed, all the predictors we developed in the past presented the same problem and this is also confirmed by all the results reported in the literature also by other groups. Following up on the reviewer comment in point 1, it is difficult to expect that the situation in devices such as ITER, DEMO and the reactor will be better, given their problems of diagnostic accessibility. Secondly, the values of the STD are so high because there are some discharges, in which the precursors manifest themselves quite early, even more than a second before the beginning of the current quench. This would give a lot of time to react and therefore it is to be considered a positive thing. This consideration is spelled out in the caption of Table III “*Table III The time intervals between the various anomalies: FT indicates the flat top and RD the ramp down phase of the discharges. Since the time intervals are never negative, large standard deviations are to be considered positive, because they indicate that in many cases the control system would have plenty of time to intervene.*” Third and more importantly, the STD is not a quantity which the algorithms rely on to make decisions about the alarms to be triggered. The statistical information was essential to set up the algorithms but the predictors do not make use of the STD to produce their outputs.

The quality of the predictors is proved by the statistics of the results reported, which show quite clearly the discriminative power of the implement logic. Indeed it has been checked a posteriori that the percentages reported in Table V and the value in Table M5.1 are correct. So the tools developed can provide quite reliably different alarms for avoidance, prevention and mitigation.

3. The method uses only a few feature sets each for a kind of instability causes. It need more evidence to show those would cover the whole plasma operation space. The shots used in the database composed on this study is not very large. I am not sure about the selection logic of this database. Can it cover the whole operational space of JET?

ANSWER: to our knowledge, the database analysed in the manuscript, which includes about 1700 discharges, is by far the largest and most comprehensive published so far for JET. It covers a couple of years of operation and it is also the most relevant, including DD, DT and TT operation. In addition, we analysed all the discharges in the campaigns, in which the diagnostic signals were available. The experimental programme in the C38-C41 campaigns was also extremely varied, since it was driven by the will to test all the major parts of JET programme in DT. Consequently, the DB analysed certainly covers the vast majority of JET operational space. Unfortunately going back further in time is not possible or relevant, because the setup of various diagnostics and codes was modified in preparation for DT. More importantly, it should be considered that the predictors are adaptive from scratch: they have managed to follow the evolution of the experimental programme for more than two years of JET operation so there is no indication that they would not be able to continue to do so in the future. The transfer of the tools to other devices has also started successfully. We have stressed further these aspects in the new version of the manuscript at the end of Section 4: *“It is also worth emphasising that the indicators devised do not show any significant dependence from the isotopic composition. This is a very important aspect, because it can potentially lead to feedback strategies, which are not too dependent on a plasma quantity, the fuel mixture, which is very delicate to control. Since the tools developed in this work are all adaptive and real time compatible, they constitute a quite encouraging package. Indeed, they have managed to follow the evolution of the experimental programme in probably the most ambitious and varied set of campaigns ever carried out on JET. Moreover, given the fact that the indicators tested are based on normalised quantities or ratios, they are expected to be easily transferrable to other devices. Indeed, the implementation of some of the proposed tools on ASDEX Upgrade, the second largest metallic tokamak in Europe, has already started and the preliminary indications are very positive.”* To conclude this point, unfortunately we have to reiterate a comment we already made to the editor. Nature Communication has already published papers on Nuclear Fusion based on a handful of discharges only (really of the order of three or four). Almost two thousand discharges are therefore well above the standards required by the journal.

4. In the method section it mentioned a NN regression model to predict the time to LM is build. I did not quite get it. what’s the input of this model? Just the features used to produce the alarm? Than why not use those NN to also output the alarm? I believe this would have better performance than just a threshold

ANSWER: the reviewer is raising a very important point that needs commenting. The method of the thresholds has been chosen for two main reasons: a) because we wanted to implement and adaptive method from scratch that could start operating after the first disruption b) because the database in the reactor will be very unbalanced since only a minimum number of disruptions will be allowed. Moreover also the algorithms of adapting the thresholds and making the

decisions to trigger alarms are implemented with networks. The quality of the obtained results is testified by the statistics reported in the various tables of the manuscript (for consistency the NN has not been used to obtain these results). The details about the NN are given in Section M4, because there is not enough space in the main text of the manuscript. We have also added a detailed list of the inputs to the network: *“The inputs are: the magnetic toroidal field (B_t), the plasma current (I_p), the dimensionless Locked Mode (LMN), the internal inductance (l_i), the electron temperature and density in the core, middle and edge regions, the electron temperature profile anomaly indicators (ETH and EC), the A_i in the four regions (Core, HF, LF and Div) and the $A_{heating}$.”*

5. The writing of this paper need revising. There are many acronyms that are missing full names. It's very confusing. In that method part, it gives a lot of citations, but also spend lots of text on talking about basic concept. But, actual useful stuff like how those features are extracted are missing especially those used for comparison.

ANSWER: we are very sorry for the difficulties encountered by the reviewer reading and interpreting our manuscript. We would take the opportunity to thank him/her again for the patience and understanding. We have now revised the wording of the manuscript with particular attention to defining the acronyms. With regard to the language, we normally proceed by asking a native speaker to polish the English once the final version has been accepted for publication. We intend to do same in this case. About the descriptions of some basic concepts, we believe they are important to make the paper accessible to a large audience. Certainly the average reader of Nature Communications does not have the same background and specialised knowledge of disruptions as the reviewer. For what concerns the features, the entire section M2 is devoted to their description; this part takes about 6 pages of the manuscript, which is already very long and heavy. Given the fact that we have used standard diagnostics and that feature extraction is not the main theme of the manuscript, we have the feeling that this subject is covered in sufficient detail (the interested reader can also make recourse to the references). Please see also answer to point 1.

6. The table V, is it a hindsight? Were the alarms raised in different time periods so they are assigned different types? It needs to be more clear and more specific on the logic of how different types of alarms were raised

ANSWER: Table V reports the results obtained applying the logic illustrated in Figure 5: *“With these objectives and in the light of the evidence just overviewed, a reasonable control strategy can be devised, whose main elements are shown in the block diagram of Figure 5. Starting from the beginning of the current quench and moving backward in time, the approach could consist of triggering mitigation actions immediately after a locked mode alarm. Following a warning due to excessive cooling of the edge, the time remaining before the beginning of the current quench is compatible only with prevention and therefore the control system should immediately activate the sequence of actions to terminate the discharge safely. In case of anomalies in the radiation patterns, as mentioned, the additional heating is to be increased in the affected regions. With sufficiently flexible additional heating schemes, such as those already deployed to avoid MHD instabilities [24,25], the regions of excessive radiation could be targeted, avoiding the radiation collapse (and consequently avoiding the disruption).”*. The predictors were run in completely real time conditions but not in feedback because closing the loop was not allowed by the control group: *“The proposed control logic has been implemented in fully compatible real time conditions, which means that all diagnostics, indicators and predictors would have worked exactly in the same way in closed feedback loop. Assuming that*

100 ms warnings for temperature hollowness and edge cooling are sufficient to undertake successful remedial action (and 10 ms warning time are enough for mitigation), for the considered database the proposed control logic provides the performances reported in Tables IV and V.”

Review to the paper

A.Murari, R.Rossi, T.Craciunescu, J.Vega and M.Gelfusa A Control Oriented Strategy of Disruption Prediction to Avoid the Configuration Collapse of Tokamak Reactors,

Results of investigation presented in the paper are very important. They related to disruption avoidance and prevention in tokamaks – one of the main barriers on the way to commercial fusion use. A new approach was developed, which allows beforehand determining the time interval remaining before an incoming disruption in real time.

So, publication of the paper is very desirable.

Volume of the paper is high (33 pages), but all chapters are quite appropriate.

ANSWER: first we would like to sincerely thank the reviewer for the time and effort spent in reading, correcting and revising our manuscript. We are also very grateful for the nice words and appreciation of our work. We found all the comments accurate and meant at improving the manuscript. We have therefore implemented practically all the requested modifications as suggested by the reviewer. We hope that the revised manuscript can be considered up to the standards of the journal but, in any case, we remain available to implement further modifications if deemed indispensable for publication.

As for reviewer, there is necessity of some corrections in the paper. Set of possible corrections is given below (page by page).

Comments and desires

p.1, it is written:

“the nature of the various forms of collapse is investigated”

Comment: yes, for collapses related to Flat Top and Ramp Down disruptions. May be add few words about disruptions at current Ramp Up stage or inform, that there is no such disruptions.

ANSWER: we have specified this point in the new version of the abstract: “*Deploying new analysis methods on thousands of JET experiments, covering the isotopic compositions from hydrogen to full tritium and including the major D-T campaign, the nature of the various forms of collapse is investigated in all phases of the discharges (including the ramp-up of the plasma current even if only a tiny number of disruptions occurs during that stage).*”

p.2, left

“repulsive coulomb barrier”

necessary

repulsive Coulomb barrier

ANSWER: many thanks for spotting the typo.

“the configuration is **plagued** by macroscopic instabilities”

better **destroyed** or **violated**

ANSWER: good point. We have modified the sentence that now reads: “*the configuration is **affected** by macroscopic instabilities*”

p.2, right, it is written

“which almost all the **internal** energy of the plasma is lost to the wall on time scales of milliseconds”.

Word **internal** here is inaccurate for my mind.

As for reviewer, not **internal**, but **plasma thermal energy** for the case of thermal quench (internal energy include thermal energy W_p and energy of plasma current inside plasma volume W_I): plasma thermal energy is $W_p = 3/2 \cdot \int (p_e + p_i) dV$; magnetic energy inside plasma volume is $W_I = L_p \cdot I_p^2 / 2$.

ANSWER: the reviewer is absolutely right. We have replaced the term “*internal*” with “*thermal*” in the new version of the manuscript.

p.3, left

in reactor **grade** devices better: in reactor **scale** devices

ANSWER: modified as proposed by the reviewer.

It is written:

“The Joint European Tokamak (JET) is the largest tokamak ever operated in the world”

Now it is not so, because of JT-60SA operation...

So, may be better:

The Joint European Tokamak (JET) is **one of** the largest tokamak ever operated in the world

ANSWER: when we originally wrote the sentence it was accurate but now the reviewer is right that this not the case anymore. We have modified the sentence as proposed by the reviewer.

p.4, left

The analysed database consists of 1735 discharges, of which 531 disruptive.

- Another numerals are in Table 1, i.e. (from Table 1: total - 1683 discharges, disruptive – 542)

It is better use the same number for total and disruptive discharges

ANSWER: the reviewer is absolutely right and we thank him/her for spotting the inconsistency. The correct values are those in the table. We have therefore corrected the main text accordingly.

p.5

Fig.2

Pulse 94199 - t = 9.2
MARFE Radiation

Necessary to add: t = 9.2 s

ANSWER: we have added the unit. Well spotted, many thanks.

p.8, left, it is written:

“All these emission events are cases of radiation runaway instabilities, because the impurities have a radiation function increasing with the inverse of the temperature”

Word “runaway” is often used in application to process of “runaway electrons evolution” during current quench.

So, may be better to write:

All these emission events are cases of radiation positive feedback instabilities, because the impurities have a radiation function increasing with the inverse of the temperature

ANSWER: good point. Modified as proposed by the reviewer.

p.8, right, it is written:

“suitable indicator proves to be the ratio of the radiated power divided by the plasma internal energy.

This quantity is much more useful than the traditional ratio of radiated power divided by input power

E_p is the plasma internal energy”

Once more: it is better to replace “internal energy” by “plasma thermal energy”

ANSWER: once more the reviewer is absolutely right. Also in this case, we have replaced the term “internal” with “thermal” in the new version of the manuscript.

p.10, right

Table III The time intervals between the various anomalies: FT indicates the flat top and RD the ramp down phase of the discharges. Since the time intervals are never negative, large standard deviations are to be considered positive, because they indicate that in many cases the control system would have plenty of time to intervene.

	Mean	Std
$\Delta t_{A,core} - \Delta t_{ETH} (all)$	167	270
$\Delta t_{A,core} - \Delta t_{ETH} (FT)$	139	435

It is necessary to indicate in Table III units of time interval (most likely in **milliseconds**?)

ANSWER: It is milliseconds indeed. Well spotted, many thanks.

p.13

Table 5, it is written:

Avoided	18.3%	18.0%	0.4%
----------------	-------	-------	------

- All must be equal to sum of FT and RD (18.3 must be equal to 18.0 + 0.4)

The same is for

Prevented	14.9%	3.9%	11.1%
------------------	-------	------	-------

 (14.9 must be equal to 3.9 + 11.1)

ANSWER: many thanks for spotting the rounding errors that we have now corrected. The tables should be coherent now.

p.13, left

The quality of such estimates are reported in **figure 6**,

Necessary:

The quality of such estimates are reported in **Figure 6**,

ANSWER: well spotted, many thanks.

p.14, right, it is written:

“From a methodological perspective, a potentially very significant improvement would consist of state aware tools, particularised for the main phases of the discharge, ramp up, flat top and ramp **of** down of the plasma current”.

May be better without **of**:

“From a methodological perspective, a potentially very significant improvement would consist of state aware tools, particularised for the main phases of the discharge: ramp up, flat top and ramp down of the plasma current”.

ANSWER: we have implemented the better wording proposed by the reviewer..

p.22, left

Written:

Assuming $T_e = T_i = T$ and deuterium fully ionised plasmas ($n_e = n_i = n$ e $\gamma = 5/3$), one obtains:

Better:

Assuming $T_e = T_i = T$ and deuterium fully ionized plasmas ($n_e = n_i = n$ e $\gamma = 5/3$), one obtains:

ANSWER: well spotted, many thanks.

Figure M3.1

instead N_e better n_e

ANSWER: good remark. We have replaced N_e with n_e in the legend of the figure.

The energy inside a certain volume V can then be calculated as:

$$E_V = \int_V \frac{3}{2} n T dV \quad (M3.3)$$

Is it good?

Recommended version:

Plasma thermal energy inside certain volume V can be calculated as:

$$E_V = \int 3nTdV \quad (M3.3)$$

Because

$$E = \int \frac{3}{2} \cdot (nT + nT) dV = \int 3nTdV$$

ANSWER: the reviewer is absolutely right. From a practical point a view, of course, a different multiplicative constant does not make any difference to the performances of the predictors. However the correct theoretical treatment is the one of the reviewer and we have implemented it. Many thanks for detecting the errors.

fitting the HRTS profiles in the region from $R = 3$ to $R = 3.9$ with a second order polynomial

Better

fitting the HRTS profiles in the region from $R = 3$ m to $R = 3.9$ m with a second order polynomial

ANSWER: we have added the unit. Well spotted, many thanks.

p.23

Table M3.1 Average difference between the estimate of the macro-pixels' energy calculated with equation M3.3

May be

Table M3.1 Average difference between the estimate of the macro-pixels' energy calculated with equation M3.4

ANSWER: well spotted, many thanks. Modified a suggested.

We can see two formulas with the same designation M4.1, i.e.

$$LM_{N,threshold} = al_i^b \quad \text{M4.1}$$

The sigmoid function to determine the probability of the plasma disrupting is:

$$P_{LM} = \frac{e^{d_{1LM}(LM_N - LM_{N,threshold}) + d_{2LM}}}{1 + e^{d_{1LM}(LM_N - LM_{N,threshold}) + d_{2LM}}} \quad \text{M4.1}$$

Better version for (M4.2):

$$p_{LM} = \frac{x_{LM}}{1 + x_{LM}} \quad (M4.2)$$

where

$$x_{LM} = e^{d_{1LM} \cdot (LM_N - LM_{N,threshold}) + d_{2LM}}$$

(all formulas must be in brackets, i.e. (M4.4), (M4.5), (M4.6), (M4.7))

$$p_{ETH} = \frac{x_{ETH}}{1 + x_{ETH}} \quad (M4.3)$$

May be better

$$p_{ETH} = \frac{x_{ETH}}{1 + x_{ETH}} \quad (M4.3)$$

where

$$x_{ETH} = e^{d_{1ETH} \cdot (ETH - ETH_{threshold}) + d_{2ETH}}$$

ANSWER: many thanks for spotting the inconsistency in the numbering of the formulas. We have corrected that and also written all formula numbers within brackets.

The plasma internal energy per unit volume can be written as:

$$u = \frac{p_i + p_e}{\gamma - 1} = \frac{n_i T_i + n_e T_e}{\gamma - 1} \quad (M3.1)$$

It's better to write The **plasma thermal energy** per unit volume...

Because internal energy usually is sum of thermal energy + magnetic energy, connected with plasma current

ANSWER: many thanks; also in this case we have substituted the adjective “*internal*” with “*thermal*” as suggested by the reviewer.

p.22, Figure M3.1

Let's check data from Fig. M3.1

We have $n_e(0) = 0.42 \cdot 10^{20} \text{ m}^{-3}$, $T_e(0) = 4 \text{ keV}$.

So electron pressure is $p(0) = n_e(0) \cdot T_e(0) = 2.7 \cdot 10^4 \text{ J} \cdot \text{m}^{-3}$. Total pressure assumed as $5.4 \cdot 10^4 \text{ J} \cdot \text{m}^{-3}$. This value of electron pressure is the same as in picture, i.e. $2.7 \cdot 10^4 \text{ J} \cdot \text{m}^{-3}$, see Fig M3.1 (right, bottom).

We see superscribe is **Energy** [$\text{J} \cdot \text{m}^{-3}$]. Here are two mistakes. Correct version of superscribe is **pressure of electrons** [$\text{J} \cdot \text{m}^{-3}$] (**pressure**, not **Energy**)

ANSWER: many, many thanks for spotting the errors that we have corrected following the reviewer indications and expressing the pressure in Pascal.

There is no Chapter Summary...

Maybe Summary is the same as Abstract. But Summary usually is more detailed, than Abstract... with some results of investigation...

ANSWER: the reviewer is right but the problem is that the space restrictions do not allow for a complete recap of all the work at the end. We have therefore substituted the word *summary* with *brief overview of the work*.

p. 30-33, References

Different way of writing names in references, for example:

[15] A. Murari

[16] Murari, A., Etc.

It is necessary to use one style...

ANSWER: thank you for spotting the issue with the format of the references. Unfortunately part of the problem is due to the fact that the journals themselves adopt different formats for their references. In any case, we have performed a first revision. If the manuscript is accepted for publication, we will iterate with the editor and implement the style recommended by the journal.

REVIEWERS' COMMENTS

Reviewer #1 (Remarks to the Author):

The authors have made substantial revisions to the manuscript in response to my concerns, significantly improving its quality. The authors' comments have addressed all my concerns. After a thorough review of the revised document, I am pleased to acknowledge the considerable enhancements that have been implemented. I also appreciate the further clarification and explanation provided regarding the STD. of the time alarm, the database analyzed in the manuscript and the results in Table V. This paper can serve as good reference for future disruption prediction research. With these improvements, I believe the revised manuscript is now suitable for publication in Nature Communications.

Reviewer #2 (Remarks to the Author):

Answers of authors are good and all mistakes and inaccuracies of the text are now in corrected form.

There is no any pretensions to the new text.

So, decision of Reviewer #2 is positive.

NCOMMS-23-47629C

. A Control Oriented Strategy of Disruption Prediction to Avoid the Configuration Collapse of Tokamak Reactors, by Murari, R.Rossi, T.Craciunescu, J.Vega and M.Gelfusa

We would really like to express our gratitude to the reviewers for recommending the publication of our manuscript.